# The Multiple Roles of Hepatitis B Virus X Protein (HBx) Dysregulated MicroRNA in Hepatitis B Virus-Associated Hepatocellular Carcinoma (HBV-HCC) and Immune Pathways

**DOI:** 10.3390/v12070746

**Published:** 2020-07-10

**Authors:** Kurt Sartorius, Leo Swadling, Ping An, Julia Makarova, Cheryl Winkler, Anil Chuturgoon, Anna Kramvis

**Affiliations:** 1Faculty of Commerce, Law and Management, University of the Witwatersrand, Johannesburg 2050, South Africa; 2Department of Public Health Medicine, School of Nursing and Public Health, University of KwaZulu-Natal, Durban 4041, South Africa; chutur@ukzn.ac.za; 3UKZN Gastrointestinal Cancer Research Centre, Durban 4041, South Africa; 4Division of Infection and Immunity, University College London, London WC1E6BT, UK; l.swadling@ucl.ac.uk; 5Basic Research Laboratory, Centre for Cancer Research, National Cancer Institute, Leidos Biomedical Research, Inc. Frederick Nat. Lab. for Cancer Research, Frederick, MD 20878, USA; ping.an@nih.gov (P.A.); winklerc@mail.nih.gov (C.W.); 6National Research University Higher School of Economics, Faculty of Biology and Biotechnology, 10100 Moscow, Russia; j-makarova@yandex.ru; 7Hepatitis Virus Diversity Research Unit, Department of Internal Medicine, School of Clinical Medicine, Faculty of Health Sciences, University of the Witwatersrand, Johannesburg 2050, South Africa; Anna.Kramvis@wits.ac.za

**Keywords:** hepatitis B virus, HBx protein, dysregulated, microRNA, hepatocellular carcinoma

## Abstract

Currently, the treatment of hepatitis B virus (HBV)-associated hepatocellular carcinoma (HCC) [HBV-HCC] relies on blunt tools that are unable to offer effective therapy for later stage pathogenesis. The potential of miRNA to treat HBV-HCC offer a more targeted approach to managing this lethal carcinoma; however, the complexity of miRNA as an ancillary regulator of the immune system remains poorly understood. This review examines the overlapping roles of HBx-dysregulated miRNA in HBV-HCC and immune pathways and seeks to demonstrate that specific miRNA response in immune cells is not independent of their expression in hepatocytes. This interplay between the two pathways may provide us with the possibility of using candidate miRNA to manipulate this interaction as a potential therapeutic option.

## 1. Background

Hepatitis B virus (HBV) infection is linked with more than 60% of all hepatocellular carcinomas (HCC) in developing countries, compared with 40% in developed countries [1], and HBV has been dubbed “the second most dangerous carcinogen after tobacco” [2,3]. Thus, HBV-associated HCC (HBV-HCC) is a leading cancer in the developing world, especially so in Africa and Asia [4]. This variant of liver carcinoma triggers a range of immune response failures that includes the dysregulation of microRNA (miRNA) [5]. miRNA provide an additional ‘layer’ of control in the immune system [6] by exerting a mild homeostatic effect on protein transcription and translation by way of suppressing complementary mRNA sequences. miRNA “see” their target by matching their nucleotide sequence to the 3′ untranslated region (UTR) of its target mRNA, whereas immune cells use selective cell surface receptors to bind with target antigens. In effect, multiple miRNA are activated in the presence of disease to collectively inhibit the mRNA expression of targeted genes in order modulate their expression. The ancillary role of miRNA, which can be described as mild suppressors acting in support of the immune system, helps to maintain homeostasis of the dynamic systems within which they operate [7].

Hepatitis B X protein (HBx)-induced dysregulation of host miRNA in the various HBV-HCC pathways [8] can contribute to the ability of HBV to evade and control the host immune system for its own purposes of replication. This modulation can result in miRNA losing their role as part of an ancillary immune system because they are commandeered to modulate host and viral expression in favor of the virus [9]. The principal purpose of this exploratory review is to illustrate the complex role of some key miRNA that are dysregulated by the HBx protein in the HBV-HCC continuum, as well as in both the innate and adaptive immune cells. In this regard, our focus is to demonstrate how the HBx protein can dysregulate miRNA in hepatocytes in HBV-HCC pathogenesis and how this can simultaneously trigger changes in the same miRNA expression in innate and adaptive immune cell pathways. This is the connection we seek to make, namely, that in HBV-HCC pathogenesis the miRNA response in immune cells is not independent of their expression in hepatocytes. We, therefore, hypothesize that in HBV-HCC pathogenesis specific HBx-dysregulated miRNA in hepatocytes also become dysregulated in immune cells because of the influence of viral infection. This review provides a platform for multiple hypotheses for future studies.

## 2. MicroRNA Expression and HBV-HCC Pathogenesis

HBV infection manifests in a range of clinical conditions including the asymptomatic carrier state, inflammation, acute or fulminant hepatitis, chronic hepatitis, and the onset of cirrhosis. Acute HBV infection only persists in 5% of adults, unlike in children where 90% of the cases do not resolve [10,11]. Moreover, if an individual develops chronic hepatitis B infection (CHB), the risk of progressing to HCC increases 100-fold if a patient is both HBsAg and HBeAg positive [12]. As viral load increases, the host immune response, triggered by viral antigens, elicits HBV-specific T-cell responses in the presence of a secondary inflammatory response, as well as increases in free radicals, interferon, tumor necrosis factor (TNF) and hepatic injury [12]. In parallel with these changes, the continuous destruction of organized extracellular matrix (ECM) and well differentiated hepatocytes results in their eventual depletion and their replacement with liver stem cells and less well organized fibrotic tissue [10]. Moreover, the integration of HBV DNA into the hepatocyte genome can trigger the oncogenic disruption of cellular genes [13] that direct apoptosis, regeneration and early senescence [14]. Oncogenic disruption leads to genomic instability that can include aberrant epigenetic change, DNA deletions, fusion transcripts *cis*/*trans*-activation, and translocations [5].

In addition to its structural proteins, HBcAg and HBsAg that form the capsid and envelope, respectively and the polymerase enzyme, HBV encodes for two non-particulate proteins, HBeAg and HBx that do not form part of the virion. The 17 kDa HBx, coded by the smallest open reading frame, *X*, is an accessory protein, which functions as a transcriptional transactivator, influencing both viral and host gene expression. Several hepatocyte signaling cascades and factors that regulate calcium, apoptosis, proliferation and the immune response can be modulated by HBx [15,16,17,18]. Unlike mammalian hepadnaviruses, the avian hepadnaviruses do not express the HBx [19]. Thus, it has been postulated that HBx may be oncogenic because hepadnavirus-associated HCC is specific to mammalian hepadnaviruses, while avian hepadnaviruses can cause chronic infection that does not progress to HCC [20,21].

The HBx protein plays an important role in the pathogenesis of viral induced HCC. This multifunctional 17 kDa protein can modulate several cellular processes directly or indirectly as a result of its interaction with the host genome. HBx integration in the host genome can influence several cellular processes including oxidative stress, cell cycle controls, apoptosis, DNA repair, as well as signal transduction, transcription and protein degradation [22,23]. HBx can also regulate the epigenetic machinery to influence access to miRNA transcription sites or influence intracellular processing by inhibiting miRNA processing steps like DROSHA/DICER machinery. The HBx protein can interact with transcription factors such as p53, nuclear factor-kappa B (NF-κB), and C-MYC, subsequently regulating miRNA expression. miRNA expression can also be modulated by HBV mRNA, which act as “sponges” to reduce expression [24].

As HBV-HCC progression proceeds from asymptomatic or acute HBV infection leading to HCC, multiple miRNA become increasingly permanently dysregulated as a result of HBV infection, inflammation [25], fibrosis [26], cirrhosis [14] and the onset of HCC [27]. The increasing level of miRNA dysregulation in the HBV-HCC continuum has been hypothesized to be a result of increased liver damage [28]. For example, one study showed 79 dysregulated miRNA in asymptomatic HBV carriers (ASC) versus 203 in CHB cases [29], while another study demonstrated an increase from 37 miRNA deregulated in healthy controls (HC) to 77 in ASCs, 101 in CHB and 135 in acute liver failure (ALF) [30].

## 3. Immune Response in HBV and the HBV-HCC Tumor Microenvironment

A few million years of ‘warfare’ between viruses and their hosts have led to the evolution of ‘clever’ viruses [31]. The first line of defense occurs in the innate immune system when interferon (IFN) molecules are produced and secreted from infected cells, in order to trigger anti-viral gene transcription and a broadly antiviral state [32]. HBV infection is characterized by a relatively delayed early innate immune response with weak induction of antiviral IFNs, as a result of poor detection of HBV and/or specific immunosuppression by viral proteins [33,34]. Consistent with findings in chimpanzees [35], HBV has been shown neither to induce nor interfere with the innate immune response in an ex vivo biopsy culture system [33] The subsequent induction of the adaptive immune system also appears delayed relative to other viral infections [31,36,37]. It is also notable that the persistent production of pro-inflammatory cytokines like IL-1β and TNF-α can also directly inhibit HBV replication [38]. Resolution of HBV infection is ultimately associated with the development of HBV-specific antibody producing B-cells and HBV-specific T-cells [39]. CD4+ and CD8+ T cell responses, especially CD4+ helper T-cells with a type 1 cytokine production, are thought to play important roles in controlling HBV infection, with CD8+ T cells capable of both suppressing viral replication and inducing lysis of infected hepatocytes [40]. Although T cell responses are induced in acute hepatitis (AH), they are significantly exhausted by their persistent exposure to HBV antigen and together with the tolerogenic environment of the liver can lead to CHB. Moreover, patients that progress to CHB demonstrate a weak or undetectable virus specific T-cell response and no detectable HBV surface antigen-specific antibody response. Viral persistence could also result from viral modulation of antigen presentation in the liver [41,42,43], for instance by suppressing pro-inflammatory cytokines [44], inducing immunosuppression that limits maturation and effective presentation of antigens by antigen presenting cells (APCs), or by presentation on non-professional or tolerizing APCs [45]. Numerous studies demonstrate that HBV also has the ability to modulate immune responses by its ability to modulate the function of dendritic cells (DCs), NK cells, T regulatory cells (Tregs) and the signaling pathways of the IFN response [40].

In CHB infection, inflammation, the development of fibrotic tissue and HBV DNA integration in the host genome, tumorigenesis can be spontaneous, or can develop over time with several characteristic changes in hepatocyte gene expression. Typically, these persistent conditions can lead to T-cell exhaustion, hyposensitivity and senescence as malignant tumors change the microenvironment [46]. The tumor microenvironment is markedly heterogeneous and comprises of various cell types including cancer-associated fibroblasts (CAFs), endothelial cells, pericytes, and immune cells. These immune cells include various types of lymphocytes, Tregs, tumor-associated macrophages (TAMs) and myeloid derived suppressor cells (MDSCs), as well as local and bone marrow-derived stromal stem and progenitor cells interspersed with surrounding ECM [47].

## 4. The Regulatory Role of miRNA in the Cancer Microenvironment

A wide range of pathological conditions in cancer are able to dysregulate miRNA modulation including the altered expression of oncogenes and tumor-suppressor genes due to chromosomal deletion or amplification, mutation and epigenetic silencing. In addition, miRNA biosynthesis can be dysregulated at multiple stages from pri-miRNA transcripts in the nucleus to mature miRNA in the cytoplasm [48]. In solid tumors, dysregulated miRNA in the tumor microenvironment can affect miRNA expression in adjacent tissue to promote carcinogenesis [49]. These mechanisms include the promotion of inflammation, angiogenesis, ECM remodeling, and immunosuppression in the neighboring tumor microenvironment [47].

Multiple miRNA in cancer cells modulate angiogenesis in the neighboring environment. For example, miR-9/-126/-135b can promote angiogenesis by indirectly amplifying the expression of VEGF-A, MERTK, IL-17 and IL-6 in the tumor microenvironment [47,50,51,52]. In this environment, miRNA are able to reprogram normal fibroblasts into CAFs [53] that are capable of promoting ECM production and increasing the secretion of cytokines and chemokines that promote tumorigenesis [54]. ECM remodeling and epithelial mesenchymal transition (EMT), are common features of carcinogenesis that are modulated by miRNA that promote this transition via the recruitment of endothelial cells to induce angiogenesis and collagen remodeling. Downregulated miR-29b, for instance, fails to modulate VEGF-A resulting in the promotion of MMP9 driven ECM remodeling [50,51,55].

## 5. HBx-dysregulated miRNA Targets in HBV-HCC and Immune Pathways

In the HBV-HCC tumor micro-environment, a range of HBx-dysregulated miRNA in hepatocytes modulate pathogenesis. Simultaneously, innate and adaptive immune cells respond to the presence of the tumor micro-environment. This response across different cell types occurs via the recognition of viral nucleic acids, viral proteins or tissue-damage and results in the activation of different families of cellular receptors [32]. This section demonstrates that the same miRNA can be dysregulated in hepatocytes in HBV-HCC pathogenesis, as well as in immune cells for a range of related reasons, e.g., to modulate pro and anti-inflammatory response [56]. This is the connection we seek to make, namely, that the miRNA response in immune cells is not independent of HBV-HCC pathogenesis in hepatocytes.

The HBx protein dysregulates multiple miRNA in HBV-HCC and these same miRNA modulate genes that potentially control innate and adaptive immune pathways in HBV-HCC (Table 1). The literature proposes four principal HCC pathways that become increasingly dysregulated as carcinogenesis progresses. These include the Retinoblastoma-Tumor Protein 53 (RB1-TP53) suppressor networks, the Phosphoinositide 3-kinase/mitogen-activated protein kinase (PI3K/MAPK) pathway, the Wingless related integration site/beta-Catenin (WNT/β-Catenin) pathway and the Janus kinase/signal transducer (JAK/STAT) pathway [57,58].

In HBV-HCC, HBx downregulated miRNA and their gene targets are illustrated in Table 1 and HBx-upregulated miRNA in Table 2. The verified gene targets in hepatocytes in HBV-HCC are listed in column 1. The verified immune gene targets of the same miRNA are separately and correspondingly shown in the second column. The immune gene targets are not all specifically identified in HBV-HCC studies and different studies include other cancer types. This second column also includes immune gene targets in both hepatocytes and leukocytes. Typically, downregulated miRNA (Table 1) fail to modulate oncogenic proteins. For example, the HBx downregulated Let-7 family members in HBV-HCC fail to modulate multiple oncogene targets like RAS/MYC/SMAD4 and WNT1.

HBx-upregulated miRNA typically repress tumor suppressor expression in HBV-HCC pathogenesis (Table 2). For example, HBx-upregulated miR-155 represses key tumor suppressors like PTEN and SOCSI in the P13/MAPK and JAK/STAT pathways, respectively. Detailed examples of the role of six key HBx-upregulated miRNA in HBV-HCC immune pathways (miR-155/-17-92/181a/-21/-29/-34) are illustrated and discussed in Section 7 (Figure 1, Figure 2, Figure 3 and Figure 4).

## 6. The Regulatory Role of miRNA in Innate and Adaptive Immune Pathways

In the absence of disease, miRNA expression constantly fluctuates in response to environmental conditions until homeostasis is restored [293]. Multiple miRNA modulate normal the innate and adaptive immune systems, first at the level of hematopoietic stem cells (HSC) and then in the differentiation and output of innate and adaptive immune cells (Table 1 and Table 2). In this context, the immune environment is influenced by an elaborate network of genes whose expression is controlled by extracellular signaling, epigenetic modifiers, transcription, splicing factors, translational protein modifiers and a constellation of miRNA [294].

This section demonstrates the regulatory role of specific miRNA in specific innate and adaptive cell pathways and contrasts with Section 5 which illustrated the target genes of HBx-dysregulated miRNA in hepatocytes in HBV-HCC pathogenesis, as well as some of their validated immune targets in both hepatocytes and immune cells.

### 6.1. miRNA and the Innate Immune System

#### 6.1.1. Granulocytes

Common myeloid progenitors (CMPs) give rise to granulocyte–monocyte progenitors (GMPs). GFi1 is a transcriptional repressor protein that controls normal myelopoiesis by regulating expression of miRNA that block granulocyte-colony stimulating factor (G-CSF)-granulopoiesis (e.g., pri-miR-21/-196b) [295]. The *BIC* gene, which is stimulated by the immune system, transcribes miR-155, which represses *SHIP*1 to promote granulocyte progenitors [231,296]. It is hypothesized that PU.1/CEBPβ promote miR-223, which represses NF1/A to promote neutrophil differentiation while PU.1/CEBPβ upregulated miR-223 can also repress MEF2C to reduce neutrophil production [297]. In addition to miR-21, granulocyte differentiation is modulated by miR-21/-223/-21/-196b/130 [294]. miR-130, for example suppresses SMAD 4 driven TGF-β1 signaling [298].

#### 6.1.2. Monocytes

It has been demonstrated that miRNA can block the transcription factors of myeloid cell development to monocytes and their differentiation into macrophages or dendritic cells. Monocytopoiesis is stimulated via colony stimulating factor receptor (CSFR), which is promoted by the expression of Runt-related transcription factor (RUNX1); also known as acute myeloid leukemia-1 (AML-1), which in turn is repressed by miR-17-92. In a feedback loop RUNX1 also suppresses miR-17-92. In monocytopoiesis it was observed that the miR-17-92 family members are downregulated resulting in the reduced modulation of RUNX1, thus promoting CSFR expression monocyte differentiation [299]. PU.1 induced miR-424 also represses NF1/A to promote monocyte differentiation. PU.1 induced miR-223 and miR-424 targets NF1/A to promote monocyte differentiation [300].

#### 6.1.3. Macrophages

In the innate immune system macrophage output is modulated by miR-155/-146a/-124/-125b/-21/-9 and Let-7e [294]. Toll-like receptor (TLR)4 signaling is increased as a result of NF-κB upregulation of miR-155, which in turn suppresses SOCS1/SHIP1, which then fail to modulate TLR4 [231,232]. The macrophage inflammatory response (TLR/NF-κB signaling) involves the upregulation of several miRNA including like miR-9/-155/-146/-147/-21 [218,232]. TLR/TNF/INF upregulation of miR-155, for instance, occurs via the activation of AP1 induced BIC transcription of this miRNA [229]. These upregulated miRNA are demonstrated to play a homeostatic role to both enhance and regulate inflammatory immune response and tissue damage. The upregulation of miR-21, for instance, suppresses the tumor suppressor PDCD4 expression which fails to modulate NF-κB signaling [301] while upregulated miR-9 provides a countermeasure by repressing NF-κB expression. TLR/RIG1 upregulation of miR-146 via NF-κB also provides a countermeasure by repressing downstream TLR inflammation activators like IRAK1, IRAK2 and TRAF6 [302]. Upregulated miR-155 can also suppress SHIP1 and SOCSI expression to reduce their negative regulation of downstream TLR signaling, thus promoting inflammatory signaling in macrophage activation [231]. However, it has been demonstrated that AKT signaling can repress miR-155 in macrophages thus indicating a negative feedback loop to fine-tune TLR signaling [303].

#### 6.1.4. Dendritic Cells (DCs)

TLR/TNF/IFN upregulated miR-155 via AP1/BIC plays a significant homeostatic role in monocytopoiesis by repressing PU.1 which activates PC-SIGN to increase pathogen cell surface uptake on DCs. Lipopolysaccharide (LPS) upregulated miR-155 modulates the TLR/Il-1 (interleukin-1) inflammation signaling pathway to regulate human monocyte-derived DCs in order to prevent excess damage [304]. DC differentiation is also modulated by miR-21/-34, which repress JAG1 and WNT1, respectively [305].

#### 6.1.5. NK Cells

NK cells express the receptor natural killer group 2, member D (NKG2D), which recognizes induced-self ligands from MHC class I-like molecules from the MIC and RAET1/ULBP families that are expressed by cells as a result of viral infection or cell transformation. NK cells are able to kill an infected or abnormal cell as a result of the engagement of NKG2D with MICA/MICB on the targeted cell. The repression of MICA/MICB by miRNA is hypothesized to reduce NKG2D engagement with NK cells thus promoting cell survival [306,307]. Several HBsAg-induced miRNA repressed the expression of MICA and MICB via targeting their 3′-untranslated regions including miR-20a, miR-93, miR-106b, miR-372, miR-373 and miR-520d [308]. The output of NK cells is influenced by miRNA like miR-181a/-150 and Let-7 [294]. Upregulated miR-181a/b, for instance, play a role in upregulating NOTCH signaling to increase NK cell output by suppressing NLK, which acts as a mediator of NOTCH expression [250].

Finally, the output of megakaryocytes, erythrocytes and other innate cells (e.g., mast cell] is modulated by miRNA like miR-10a/-150/-144/-451/-221/-222/-223 [294]. miR-221/-222 suppress p27 to influence mast cell proliferation [264] while miR-144/-451 suppresses RAC1 and ETS2 to influence megakaryocyte erythrocyte progenitor (MEP) output and differentiation, respectively [309].

### 6.2. miRNA and the Adaptive Immune System

Lymphopoiesis is modulated at various stages by miRNA from common lymphoid progenitors (CLP) to the final output of mature T and B cells.

#### 6.2.1. T-Cells

Proliferating T-cells have been shown to synthesize mRNA with shorter 3′ binding sites thus potentially rendering them less sensitive to miRNA induced silencing than resting T-cells [310]. DICER deficiency, for instance, has been demonstrated to influence aberrant T-cell differentiation [311]; however, two specific miRNA have been shown to play a specific role in T-cell development, namely, miR-17-92 and miR-181a. miR-17-92 members target BIM/PTEN to promote cell survival in the double negative (DN) to the double positive (DP) stage [312], while miR-181a targets DUSP5/DUSP6/SHP2/PTPN22 in the DP to single positive (SP) stage to increase TCR signaling and influences the antigen recognition sensitivity of mature T-cells [249]. In the Th1/2 differentiation stage miR-155 expression is thought to promote differentiation into Th1 cells as a result of targeting MAF [228,229]. This BIC encoded miRNA also represses SOCSI that, in turn, represses Treg generation to regulate autoimmune response [313,314]. miR-326 regulates Th-17 differentiation via the repression of ETS1 [315].

The activation and proliferation of T-cells is further influenced by miR-181a stimulated TCR signaling; however, a negative feedback loop represses the output of T-cells because miR-181a modulates CD69 led activation of T-cell output. CD69 is further repressed when TCR led induction of miR-17-92 family members targets this protein’s expression, thus providing an additional check point for controlling T-cell output [244]. Multiple miRNA influence the differentiation and output of Th17 cells including miR-155/-21/-301/-326/206 [294]. miR-21, for instance, suppresses SMAD7 thus influencing TGFβ led signaling to promote Th17 differentiation [166], while miR-155 influences Th17 differentiation by suppressing SOCSI [238].

#### 6.2.2. B-Cells

B-cell development in the bone marrow is controlled by the commitment of progenitor cells to the B-cell lineage as a result of the activation of transcription factor networks, as well as V(D)J recombination and the selection of antigen receptors [65]. In the early stages of development, the overexpression of miR-181 skewed leukopoiesis towards the development of B-cells at the expense of T-cells by repressing DUSP5, DUSP6, SHP2 and PTPN22 [316], while miR-150 can repress C-MYB to reduce Pro-B cell development [317,318] and miR-17-92 absence has been demonstrated in DICER deficient Pre-B cells where this miRNA fails to repress BIM thus promoting its pro-apoptotic effect and preventing Pre-B cell development [243].

Mature B-cell differentiation is modulated by miR-155, which targets AID thus regulating GC B-cell versus Marginal zero B-cell development. This crucial miRNA also targets PU.I to block GC B-cell to plasma cell transition thereby modulating B-cell differentiation into memory cells or plasma cells. B-cells, that are miR-155-deficient, can have a defective humoral response to T-cell-dependent antigenic stimulation because of an impaired antibody class switching and differentiation into plasma cells [228,229,319]. In the adaptive immune system B-cell development is modulated by miR-181/-150/-212/-132/-17-92/-34a/-21/-148/-125b/146a/155 [294]. Upregulated members of the miR-17-92 family increase pro-B-cell to pre-B-cell transition by suppressing BIM [243], while p53 upregulated miR-34a has been reported to reduce pro-B-cell to pre-B-cell transition because of the suppression of the FOXP1 oncogene [193].

In the next section, we demonstrate that the same HBx-dysregulated miRNA in HBV-HCC in hepatocytes can be interdependently activated in the innate and adaptive cell pathways.

## 7. HBx-Dysregulated miRNA in HBV-HCC and in Immune Pathways

Upregulated miRNA typically reduce tumor suppressor expression in the four key HCC cancer pathways, namely, the P13K/MAPK, WNT/β-Catenin, TP53 and JAK/STAT pathways [57]. Examples of the complexity of the interlocking roles of miRNA in HCC pathogenesis and modulation of the host immune system are illustrated in Figure 1, Figure 2, Figure 3 and Figure 4. HBV infection can dysregulate multiple miRNA in order to ‘cleverly’ modulate the host immune response to promote its own replication and/or viral persistence. In this section, we present a few examples of HBx-dysregulated miRNA that are reported in both HBV-HCC and hematopoiesis. Many questions remain with respect to the influence of HBV infection in HBV-HCC, hematopoiesis and the role of the HBx protein. We will now review in more detail the literature on six well-characterized miRNA (miR-155, mir-17-92, miR-181a, miR-21, miR-29a/b and miR34) that are dysregulated in HBV-HCC and the diverse roles they play in lymphocyte subsets. These specific miRNA were also selected because they are all play a modulating role in highly researched cancers like those of the breast, lung and colon, as well as in HBV-HCC and in cancer-related immunology studies focusing on leukopoiesis. It is important to highlight in Figure 1, Figure 2, Figure 3 and Figure 4 that the proposed miRNA immune pathways have been demonstrated in multiple cancers, including HCC, but to some extent contain a hypothetical element. This is because miRNA dysregulation in the immune pathways can be caused by factors in addition to HBx, such as TLR/NF-κB signaling, inflammation, APCs and the expression in each of the immune pathways in the figures could be dynamic in an HBV-HCC context.

### 7.1. HBx-Dysregulated MiR-155 in HBV-HCC and in Immune Pathways

MiR-155 is a multifunctional miRNA that plays an important ancillary regulatory role in the immune system in response to disease [320]. This miRNA is expressed in a variety of immune cell types, including B cells, T cells, macrophages, DCs, and progenitor/stem cell populations. Normally, miR-155 is found a) ligands, and inflammatory cytokines, which rapidly increase miR-155 expression [65]. This miRNA has an important role in regulating cytokine production, inflammation, as well as in modulating myeloid and lymphoid differentiation [228]. In the immune system, miR-155 is unique in its ability to shape the transcriptome of activated myeloid and lymphoid cells [321].

As in breast, lung and colon cancer [322,323,324], miR-155 is frequently dysregulated in HBV-HCC pathogenesis (see Table 2). In the P13K/MAPK pathway, HBx-upregulated miR-155 represses PTEN to promote downstream AKT/MTOR signaling and epithelial to mesenchymal transition in HBV-HCC progression [226,325]. In the WNT/β-Catenin pathway, this HBx-dysregulated miRNA represses the APC/GSK3 destruction complex to release β-Catenin-directed transcription of oncogenic proteins like C-MYC [24,326]. miR-155 also plays a role in JAK/STAT pathway by repressing the SOCSI tumor suppressor to increase downstream signaling for the transcription of *CCND*1 and *C-MYC* to promote HCC cell proliferation [327,328]. In the TP53 pathway, miR-155 represses SOX6 to reduce its role in upregulating tumor suppressor expression of p21/Waf1/cip1 thus promoting reduced cell cycle controls and promoting HCC proliferation [8,224]. This HBx-upregulated miRNA also represses HBV replication by modulating CCAAT/enhancer-binding protein (C/EBP) protein that activates the Enhancer 11/basal core promoter [326].

#### 7.1.1. Innate Immune System

MiR-155 modulates a range of pro- and anti-inflammatory responses in the innate immune system [56,230]. This BIC transcribed miRNA plays a major role in the modulation of NF-κB driven induced myelopoiesis by targeting IRAK1/TRAF6 and SHIP1/SOCS1 respectively [237,238,329]. SHIP1 is a primary target of miR-155 and its repression influences an increase in granulocyte/monocyte cell populations and a reduction in lymphocyte numbers [231,296]. It was observed that reduced levels of SHIP1 in the hematopoietic system induce myeloproliferative disorders [231]. This miRNA also targets CSFR, which may influence myeloid differentiation [65].

#### 7.1.2. Macrophages

It was observed in macrophages that RNA virus infection can induce miR-155 expression via the TLR/MyD88/JNK/NF-κB dependent pathway to promote type I IFN signaling, thus suppressing viral replication, possibly to promote evasion and survival objectives. Furthermore, SOCS1, a canonical negative regulator of type I IFN signaling, is targeted by miR-155 in macrophages, and SOCS1 knockdown mediates the enhancing effect of miR-155 on type I IFN-mediated antiviral response [330,331]. TLR/TNF/IFN upregulation of miR-155, for instance, occurs via the activation of AP1 induced BIC transcription of this miRNA [229]. Upregulated miR-155 can also suppresses SHIP1 and SOCSI expression to reduce their negative regulation of downstream TLR signaling thus promoting inflammatory signaling in macrophage activation [231]. However, it has been demonstrated that AKT signaling can repress miR-155 in macrophages thus indicating a negative feedback loop to fine-tune TLR signaling [303]. The dysregulation of the SOCS-1 function as a tumor suppressor is common in HCC pathogenesis and the HBx mediated upregulation of miR-155 is a contributing factor in HBV-HCC [327,328].

#### 7.1.3. Dendritic Cells (DCs)

TLR/TNF/IFN upregulated miR-155 via AP1/BIC plays a significant homeostatic role in monocytopoiesis by repressing PU.1, which activates PC-SIGN to increase pathogen cell surface uptake on DCs. LPS upregulated miR-155 modulates the TLR/IL-1 (interleukin-1) inflammation signaling pathway to regulate human monocyte-derived dendritic cells in order to ensure excess damage does not occur [304]. Decreased DC-SIGN expression in HCC is related to poor prognosis and PU.I has been identified as a metastasis suppressor possibly relating to the impairment of the antigen presenting capabilities of APCs [332]. TLRs, as well as the nuclear factor (NF)-*κ*B, and JNK pathways are critical regulators for the production of the cytokines associated with tumor promotion. The cross-talk between an inflammatory cell and a neoplastic cell, which is instigated by the activation of NF-*κ*B and JNKs, is critical for tumor organization [333].

#### 7.1.4. Adaptive Immune System

##### T-Cell

MiR-155 especially modulates T helper cell differentiation and the germinal center reaction to produce an optimal T cell dependent antibody response [229]. In the Th1/2 differentiation stage miR-155 expression is thought to promote differentiation into Th1 cells as a result of targeting c-Maf [228,229] and an elevated Th17 to Th1 ratio has been associated with tumor progression in HBV-HCC [334]. miR-155 in Th17 cells can also trigger autoimmune inflammation through a signaling network by targeting the Ets1/IL-23/IL-23R pathway [237].

This BIC encoded miRNA also represses SOCSI that, in turn, represses Treg generation to regulate autoimmune response [313,314]. Upregulated miR-155 enhanced Treg and Th17 cells differentiation and IL-17A production by targeting SOCS1 [238]. A meta-analysis indicated that the increased expression of Tregs has been associated with the promotion of HCC. This study also demonstrated that Treg levels in the HCC tumor microenvironment were significantly higher than in normal surrounding tissue [335]. Conversely, Fox3p directly targets miR-155 resulting in a reduction in Tregs [227]. This miRNA also modulates IFNγ expression through a mechanism involving repression of Ship1 showing the critical roles for miRNA in the reciprocal regulation of CD4+ and CD8+ hematopoiesis [221]. miR-155 also plays a role in the generation of exhausted dysfunctional T cells during chronic antigen exposure. Fosl2 antagonism of miR-155 reduced could even reduce T cell exhaustion during chronic viral infection [336].

##### B-Cell

Mature B-cell differentiation is modulated by miR-155, which targets AID thus regulating germinal center (GC) B-cell versus marginal zone B-cell development. This crucial miRNA also targets PU.I to block GC B-cell to Plasma cell transition thereby modulating GC B-cell differentiation into memory cells or plasma cells. This miRNA, therefore, plays an important role in regulating the germinal center reaction in part by directly down-regulating AID expression [236].It has also been demonstrated that miR-155 modulates the generation of class switched B-cells by acting as a suppressor of the AID enzyme, which is essential for class switch recombination (CSR). Modulating miR-155 expression demonstrates that upregulated miR-155 will reduce generation of CSR and downregulated miR-155 will increase the net effect [6]. In B-cells with miR-155 deficiency, it has been noted that there is defective antibody class switching and differentiation into plasma cells resulting in reduced T cell expression that is dependent on antigenic stimulation [228,229,319]. Overexpression of miR-155 is linked to many cancers of B-cell origin [321].

### 7.2. HBx-Dysregulated miR-17-92 Family in HBV-HCC and in Immune Pathways

Dysregulated miR-17-92 is widely reported in lung, colorectal and breast cancer [337,338,339]. In HBV-HCC the HBx protein can transactivate C-MYC to upregulate miR-17-92 family members. Conversely, the miR-17-92 family members can counter regulate C-MYC expression [240]. In the P13K/MAPK pathway upregulated mIR-17-92 suppression of PTEN which then fails to modulate P13K/MTOR signaling resulting in an increase in HCC carcinogenesis [340,341]. The suppression of PTEN tumor suppressor has been widely linked to HCC [342].

In the TP53 pathways, upregulated miR-17-92 family repress E2F1 and p21/p27 and p57 cell cycle controls to upregulate cell proliferation and promote cell survival [239,240,241].

#### 7.2.1. Innate Immune System

##### Monocytes

Monocyte production is stimulated by CSFR, which is promoted by RUNX1 expression that in turn is repressed by miR-17-92. In a feedback loop RUNX1 also suppresses miR-17-92. In monocytopoiesis it was observed that miR-17-92 family members can be downregulated resulting in the reduced modulation of RUNX1 thus promoting CSFR expression and monocyte differentiation [299]. Downregulation of RUNX1 is a feature in HCC [343], conversely the upregulation of RUNX1 has been linked to a reduction in HCC because RUNX1 suppresses VEGFA leading to reduced proliferation and migration [344]. CSFR stimulation is linked to increased macrophage activity, inflammation, tissue remodelling and HCC [345,346].

#### 7.2.2. Adaptive Immune System

##### T-Cells

The upregulated expression of miR-17-92 miRNA can repress the tumor suppressor PTEN and the pro-apoptotic protein BIM to promote lymphoproliferative disorders and autoimmunity [312]. Suppression of PTEN by miR-17-92 also promotes Th1 response versus Treg generation [245]. miR-17-92 members play a key role in T-cell development by targeting BIM/PTEN to promote cell survival in the DN to DP stage [312]. TCR led induction of miR-17-92 family members also target CD69 to control CD69 expression, which provides an additional check point for controlling T-cell output [244]. BIM and CD69 are responsible for the termination of acute inflammatory response by repressing excess T-cell production. Death following re-stimulation of the TCR, as occurs during activation-induced cell death, is known to depend on the CD95–CD95 ligand pathway [347], which is an early leucocyte activating molecule in chronic inflammation [348].

##### B-Cells

The absence of miR-17-92 leads to increased levels of the pro-apoptotic protein BIM and inhibits B cell development at the pro-B to pre-B transition [243]. Upregulated miR-17-92 can also suppress BIM to promote B-cell development [349]. The miR-17-92 family modulates the migration of CD4^+^ T cells into B cell follicles by repressing PHLPP2, which induces the co-stimulator ICOS and kinase PI(3)K promotion in T-follicular helper (T_FH_) cell differentiation [242].

### 7.3. HBx-Dysregulated MiR-181a in HBV-HCC and in Immune Pathways

Dysregulation of miR-181a occurs in breast, lung and colorectal cancer studies [350,351,352]. In the TP53 cancer pathway HBx upregulation of miR-181a contributes to increasing HCC proliferation by downregulated E2F5 expression [247], as well as facilitating HCC survival by suppressing FAS to promote an anti-apoptotic response [246]. This miRNA can also inhibit autophagy in HCC by targeting autophagy-related gene 5 (Atg5), resulting in decreased apoptosis of HCC cells and increased tumor growth [353]. In the P13k/MAPK pathway HBx-upregulated miR-181a suppresses PTEN to increase AKT/MTOR signaling that stimulates HCC progression [354]. Interestingly, miR-181a expression is also elevated by WNT/β-Catenin signaling [355].

#### 7.3.1. Innate Immune System

miR-181a has emerged as an important homeostatic agent to modulate inflammation in HBV-HCC and immune pathways.

#### 7.3.2. Monocytes and Macrophages

Upregulated miR-181a regulates inflammatory responses by directly targeting the 3′-UTR of IL-1a and down-regulating IL-1a levels. Thus, miR-181 and IL-1a have opposite expression levels in monocytes and macrophages in the inflammatory state during HBV-HCC promoting an anti-inflammatory response [356].

#### 7.3.3. Dendritic Cells

MiR-181a can repress the inflammatory response in DCs cells by targeting FOS. It has also been demonstrated that the expression of FOS is elevated in human hepatoma compared with adjacent tissues [357]. It is, therefore, hypothesized that this HBx-upregulated miRNA modulates an anti-inflammatory response in DCs in HBV-HCC by targeting FOS. In addition, miR-181a also modulates an anti-inflammatory response by targeting Il-6 and TNFα whose elevation is noted in HCC. This miRNA, therefore, attenuates the carcinogenic properties of these two proteins in HCC [358,359]. A further role of this miRNA in DCs regulates ubiquitination targeting FOS. Interestingly, the ubiquitin C (UBC) gene has been cited for its role HCC pathogenesis [360].

#### 7.3.4. NK Cells

Upregulated miR-181a plays a role in promoting NK cell output by upregulating NOTCH signaling. The upregulation of NOTCH signaling occurs because miR-181a represses NLK, which, in turn, represses NOTCH expression [250]. NOTCH signaling is activated in HCC and induces tumor formation, implying that the suppression of NOTCH signaling will attenuate HCC progression [361].

#### 7.3.5. Adaptive Immune System

##### T-Cell

MiR-181a plays a role in the activation and proliferation of T-cells by stimulating TCR, as well as modulating both T-cell and B-cell differentiation. This miRNA augments TCR signaling by repressing TCR antagonists like DUSP5/DUSP6/SHP2/PTPN22 [249]. This miRNA plays a specific role in the development of adult T-cells by modulating the DN to DP transition by modulating PTPN22/SHP2/DUSP6. A negative feedback loop; however, modulates the output of T-cells because miR-181a also suppresses CD69 led activation of T-cell output. CD69 expression can terminate acute inflammatory response by repressing excess T-cell production [249].

##### B-Cell

In early stage development, the overexpression of miR-181 skewed haematopoiesis towards the development of B-cells at the expense of T-cells by repressing DUSP5, DUSP6, SHP2 and PTPN22 [316].

### 7.4. HBx-Dysregulated MiR-21 in HBV-HCC and in Immune Pathways

MiR-21 is reported as an oncogenic miRNA in lung, colorectal and breast cancer [322,362,363]. HBx-dysregulated miRNA modulates the P13K/MAPK and WNT/β-Catenin pathways in HCC. In the P13K/MAPK upregulated miR-21 represses PTEN to reduce modulation of AKT/MTOR signaling contributing to increased HCC proliferation [354]. In the WNT/β-Catenin pathway this upregulated miRNA promotes carcinogenesis via two sub-pathways. Firstly, it can suppress DCC6 thus preventing its modulating of WNT signaling. Secondly, this miRNA represses PDCD4, which then fails to repress SNAIL suppression of Cadherin expression which contributes to migration and increased β-Catenin expression [253,255,256].

#### 7.4.1. Innate Immune System

HBx-upregulated miR-21 controls a balance of pro and anti-inflammatory immune responses and elevated miR-21 levels are a marker of immune cell activation [165].

#### 7.4.2. Macrophages

The upregulation of miR-21 promotes a pro-inflammatory response in macrophages by repressing the tumor suppressor PDCD4 expression which then fails to modulate NF-κB signaling [301]. Conversely, the miR-21 downregulation of PDCD4 reduces its repression of IL-10, thus promoting an anti-inflammatory response [301]. It has been proposed that the poor immune response in tumor activated macrophages may be explained by increased levels of IL-10 [165,301].

#### 7.4.3. Dendritic Cells

MiR-21 modulates monocyte-derived dendritic cell (MDDC) differentiation by repressing JAG1 and WNT1 [305].

#### 7.4.4. Adaptive Immune System

##### T-Cells

HBx-upregulated miR-21 can promote Th17 differentiation by targeting and depleting SMAD-7, a negative regulator of TGF-β signaling [166]. miR-21, for instance, suppresses SMAD7 thus influencing TGFβ led signaling to promote Th17 differentiation [166]. Upregulated miR-21 also represses IL-12, which acts as a strong inducer of Th1 responses thus reducing IFNγ production and a reduction in Th1:Th2 ratio in T-cell production [364] demonstrating that this interaction supports the notion that miR-21 controls the balance of pro- and anti-inflammatory responses [165]. Upregulated miR-21 also suppresses production of the potent antiviral cytokine IFN by repressing MYD88/IRAK [164]. Conversely, miR-21 expression can promote NFκB activation and TNF-α and IFNγ production in activated T-cells clearly acting to induce inflammation on recognition of transformed tumor-cells [365].

### 7.5. Other Key MIR and Immune Pathways Dysregulated by HBx in HBV-HCC 

Dysregulation of miR-29 occurs in lung, breast and colon cancer [366,367,368]. HBx-upregulated miR-29a/b is widely reported in miRNA modulation of HBV-HCC pathogenesis by targeting genes like PTEN/PI3K/AKT/MMP-2 in HBV-HCC, thus contributing towards the promotion of cell migration and invasion [127,276]. The miR-29a/b cluster plays a crucial role in the thymic production of T-cells, T-cell differentiation and B-cell oncogenic transformation [277]. In the presence of infection, this family of miRNA modulates type 1 IFN signaling and T-BET/EOMES expression promoting Th1 CD4+ differentiation over Th2 differentiation. Downregulated miR-29a/b, for instance, fails to block type 1 IFN/T-BET/EOMES thus promoting Th1 CD4+ differentiation. Conversely, upregulated miR-29a/b blocks type 1 IFN/T-BET/EOMES to promote equal expression of Th1 and Th2. However, in HBV-HCC miR-29a/b is upregulated by the HBx protein, suggesting a viral intervention to promote a balanced expression of Th1 and Th2. A similar role is played by miR-29a/b when this miRNA is downregulated by intracellular bacteria and fails to modulate type 1 IFN resulting in an imbalance of the production of CD8+ T-cell [277].

Dysregulation of miR-34 has been reported in colon, breast and lung cancer [369,370,371]. Another HBx-dysregulated miRNA, miR-34, acts across the entire HBV-HCC continuum and is upregulated in early HBV infection/inflammation [24] and in HBV induced fibrosis [372]. However, miR-34 is widely reported as relatively downregulated in HBV-HC playing a role in the modulation of metastasis, growth and apoptosis [188]. The HBx protein can repress p53 stimulated miR-34 in hepatocytes leading to an upregulation of macrophage-derived chemokine (CCL22) stimulated regulatory T-cells (Tregs). Tregs, in turn, can block effector T-cells thus allowing HBV expression to increase [24,187]. Upregulated p53 induced miR-34a is also reported to suppress FOXP1 resulting in the inhibition of pro-B cell to pre-B cell transition [193]; however, if HBx suppresses p53 led stimulation of miR-34a [24] then the consequent will be different.

## 8. Conclusions

In this extensive review we have attempted to bring together studies that have shown the complex interlinking roles of miRNA in HBV-HCC pathogenesis and the immune response, both innate and adaptive. Moreover, from the literature it is evident that nearly all HBx-dysregulated miRNA in HBV-HCC can additionally act on multiple immune targets (Table 1 and Table 2). Using four key miRNA as an illustration, it is clear that there is simultaneous modulation of central pathways, namely, the principal HBV-HCC cancer pathways and those of the innate and adaptive immune systems (Figure 1, Figure 2, Figure 3 and Figure 4). We, therefore, hypothesize that the same specific miRNA that are dysregulated in hepatocytes during HBV-HCC pathogenesis can become simultaneously and interdependently dysregulated in immune cells and vice versa. The four representative miRNA selected primarily demonstrate how they modulate HBV replication and oncogene or tumor suppressor expression in HBV-HCC pathogenesis while simultaneously modulating the proliferation and differentiation of leucocytes in the innate and adaptive immune systems. This interplay between the two pathways may provide us with the possibility of using candidate miRNA to manipulate this interaction as a potential therapeutic option.

Multiple miRNA target the same genes and post-transcriptional gene silencing of translation is a collective effort. Even then it is likely miRNA only exert a mild secondary influence on mRNA stability and translation in response to the stochastic nature of gene expression and changing environmental influences [7]. Furthermore, small tumors (<0.5 cm) would be unable by themselves to alter the level of extracellular miRNA in sera and the explanation for dysregulated miRNA in early stage carcinogenesis would likely be as a result of general immune responses [373]. In vivo results also indicate that most RNA-based therapies are compromised by non-specific organ bio-distribution, reticuloendothelial system (RES) clearance, and endolysosomal trafficking [374]. Increasingly, future studies will need to consider the selection of sub-populations of extracellular vesicles that facilitate small RNA messaging. Emerging research indicates that only certain types of encapsulated miRNA play a role in cell-cell signaling and others may not. Exosomes, for instance, appear to transport miRNA that promote paracrine communication [375,376,377] and nanotechnology can be used to deliver chemically modified miRNA to cancer cells [378,379].

This rather simplistic account cannot illustrate the full extent of the dynamic, complex and multi-dimensional role of each miRNA in varying HBV-HCC cases either with respect to the varying degrees of expression in each pathway or the degree to which HBV-HCC pathogenesis can be modulated. However, the demonstration of these interrelationships will allow each of these potential interactions to be treated as hypotheses that need to be tested individually. Although miRNA hold promise as therapeutic agents in various cancers including HBV-associated HCC, this field of study remains a work in progress that is yet to be fully exploited [380].

## Figures and Tables

**Figure 1 viruses-12-00746-f001:**
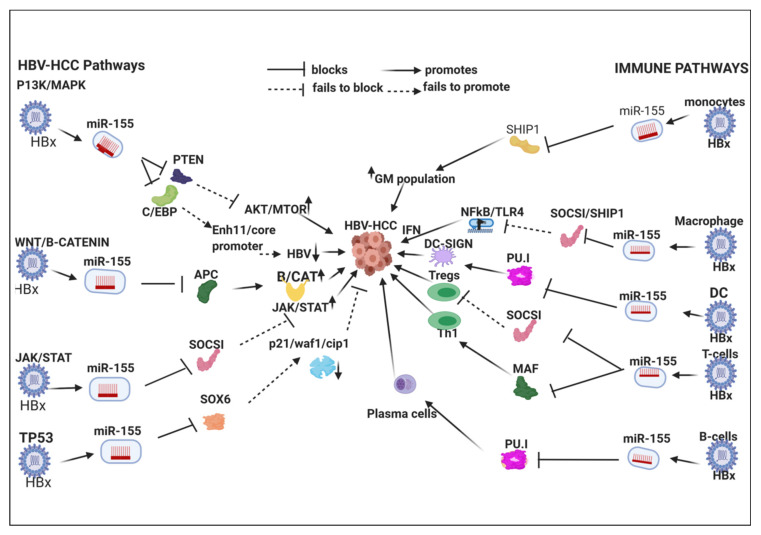
HBx induced MiR-155 in HBV-HCC immune pathways: This HBx-upregulated onco-miRNA promotes MTOR signaling and EMT in the P13K/MAPK pathway; it promotes β-Catenin expression to promote oncogenic proteins like C-MYC in the WNT/β-Catenin pathway; it subdues SOCSI suppression of JAK/STAT signaling to promote oncogenic proteins like C-MYC and CCND1 in the JAK/STAT pathway and it reduces expression of tumor suppressors like p21/waf1/cip 1 to promote cell proliferation in the TP53 pathways. This upregulated miRNA, however, reduces HBV replication by repressing C/EBP promotion of ENH11/core promoter. In the immune pathway, this miRNA influences granulocyte/monocyte populations via repressing SHIP1; it represses SHIP1/SOCS1 to promote NF-κB/TLR induction of macrophages; in DCs this miRNA represses PU.1 induction of DC-SIGN to reduce pathogen cell surface uptake; in T-cell synthesis this miRNA can repress SOCS1 to promote Treg production, it can promote Th1:Th2 ratio by repressing C-MAF; in B-cells this miRNA represses PU.1 to promote GC differentiation into memory or plasma cells.

**Figure 2 viruses-12-00746-f002:**
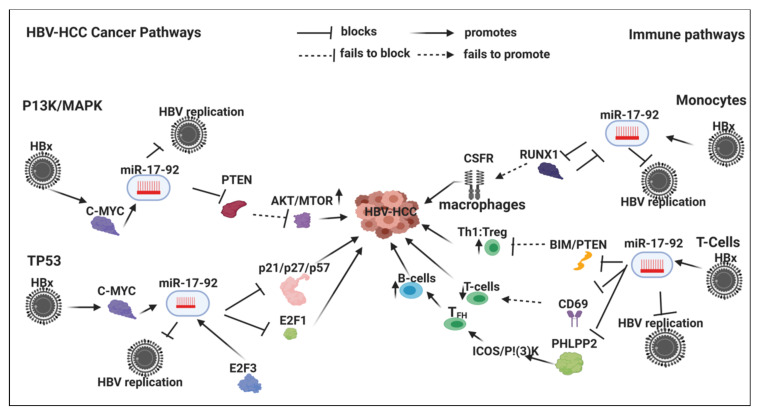
HBx-upregulated MiR-17-92 family in HBV-HCC immune pathways: This HBx-dysregulated miRNA family (via C-MYC) promotes HBV-HCC progression in the P13K/MAPK by repressing PTEN to upregulate MTOR signaling; in the TP53 pathways it can increase cell proliferation by repressing p21/p27/p57 and E2F1 cell cycle control primarily by promoting MTOR signaling and blocking cell cycle controls. Upregulated family can also repress HBV replication. In monopoiesis, upregulated family members can increase macrophage development via repressing RUNX1 to promote CSFR stimulation; in T-cells upregulated family members can repress PTEN/BIM to increase Th1 versus Treg expression; conversely it can repress CD69 to modulate T-cell output; B-cell output is increased when upregulated family members repress PHLPP2 to promote ICOS/P1(3)K stimulation of T_FH_ induced B-cell response.

**Figure 3 viruses-12-00746-f003:**
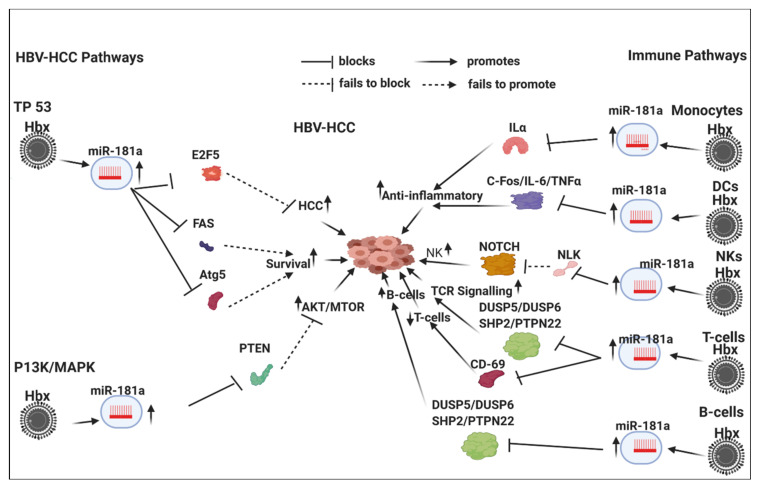
HBx-induced mIR-181a in HBV-HCC immune pathways: This HBx-upregulated miRNA promotes cell proliferation in the TP53 pathway by repressing cell cycle controls like E2F5 and it exerts an anti-apoptotic influence by repressing FAS/ATG5 to promote cell survival; in the P13K/MAPK this upregulated miRNA also promotes carcinogenesis by promoting MTOR signaling as a result of repressing PTEN. In the innate immune pathways, this miRNA promotes an anti-inflammotory response by repressing Ilα and C-FOS/IL-6/TNFα in monocytes and DCs respectively; in NKs this miRNA upregulates NKs by repressing NLK, which then fails to repress NOTCH induced induction of NKs; in T-cells this miRNA represses DUSP5/6/SHP2/PTPN22 to increase TCR signaling induced stimulation of T-cells; however, this upregulated miRNA also represses T-cell production by reducing CD69 expression; in early stage leukopiesis this miRNA can also promote B-cell to T-cell differentiation in favor of B-cells by repressing DUSP5/6/SHP2/PTPN22.

**Figure 4 viruses-12-00746-f004:**
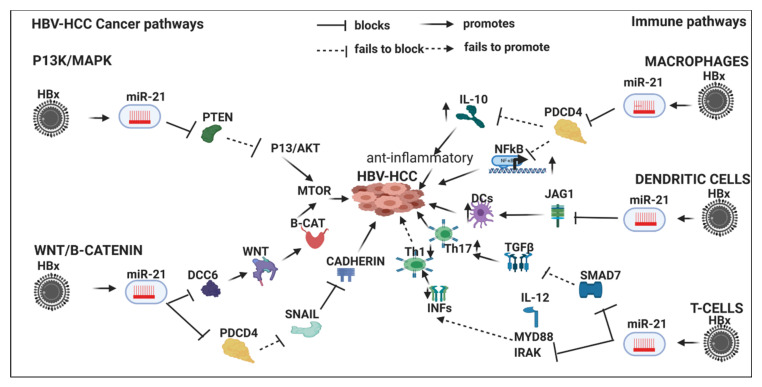
HBx-induced MiR-21 in HBV-HCC immune pathways: In the P13K/MAPK pathway this HBx-upregulated miRNA also promotes HCC by upregulating mTOR signaling via repressing PTEN; in the WNT/β-Catenin pathway, it promotes the onco-protein β-Catenin by regulating a suppressor of WNT signaling, as well as by repressing CADHEREN via reducing PDCD4 modulation of SNAIL which acts as a repressor of CADHEREN. In macrophages, this upregulated miRNA exerts a pro and anti-inflammatory influence by repressing PDCD4. In the first case, the repression of PDCD4 reduces its own repressive of pro-inflammatory NF-κB led signaling, in the second case the repression of PDCD4 stimulates the upregulation of the anti-inflammatory IL-10; this upregulated miRNA can increase DC output by repressing JAG1; in T-cells, this miRNA can promote Th17 expression by suppressing SMAD7, which is a negative regulator of TGFβ, as well as reduce Th1:Th2 ratio by targeting IL-12 induction of INFs to promote Th1.

**Table 1 viruses-12-00746-t001:** HBx-downregulated miRNA targets in HBV-HCC and immune pathways.

miR	HBV-HCC Target (Hepatocytes)	Immune Target (Hepatocytes/Leukocytes)	Reference
let-7/miR-98	STAT3/RAS/HMGA2/MYC/ IL-6/IL-10/TLR-4/COL1A2/NGF/BCL-XL/BCL-2/MCL-1	MYC/STAT3/IFN-b/RAS/TLR4/BCL-XL/SMAD2/SMAD4/APC2/WNT1/HMGA2/PLZF/IFNγ/IL-4/IL-17/LIN28B/IGF2BP1/NF2	[59,60,61,62,63,64,65,66,67,68,69,70,71,72]
miR-1	EDN1/PI3K/AKT/HDAC4/MET	HDAC4/E2F5/HSP60/HSP70/KCNJ2/GJA1	[73,74,75,76]
miR-101	GSTP1/FOS/EZH2/MCL-1/DNMT3A/RASSF1/PRDM2	ICOS (naïve T-cells)/MCL-1	[9,77,78,79,80,81,82]
miR-101-3p	ND, RAP1B/MCL-1,SOX9	ICOS/MCL-1	[83,84]
miR-122	CTNNB1	SOCS3/IFN/IP-10/BCL-W	[24,85,86,87,88]
	CCNG1 modulated p53/GLD2		[89,90,91,92]
	NDRG3/GALNT10/CCNG1/PTTG1		[93,94,95,96]
	PBF/ADAM10/CCNG1/Igf1R/ADAM 17/BCL-W/NDRG3		[24,89,96,97,98,99]
miR-124	STAT3 and PIK3CA	STAT3/TRAF6/CYCLIND3/BM11	[100,101,102,103]
miR-125b	SMAD2/4/Sirtuin7/SUV39H1/ LIN28 B/PIGF/BCL-2/MCL-1	PRDM1/IRF4/TNFα/BCL-2/MCL-1/LIN28/IRF4/KLF13/BMF/BCL-2/SMAD2/SMAD4/APC2/WNT1/BLIMP1/IRF4/BMF/KLF13/TRP53INPI/LIN28A/IRF4	[24,67,68,69,104,105,106,107,108,109,110,111,112]
miR-132	AKT	p300/IRAK4/FOXO3/SOX4/	[113,114,115,116,117]
miR-136	AEG-1	RIG-1/NF-κB	[118,119,120]
miR-138	CCND3/CDK4/6	CTLA-4/PD-1/PD-L1	[121,122,123]
miR-139-5p	ZEB1/2	IL-4/IFN-γ	[24,124,125,126]
miR-145	MAP3K/CUL5/HDAC2/ADAM17	IFN-b/TIRAP/TRAF6	[63,127,128,129,130]
miR-148a	HPIP/AKT/ERK/FOXO4/ATF5/ MTOR/MET/ACVR1	CaMKIIα/KIT/MET/SIPI/BACH/PTEN/BIM/GADD45/	[24,131,132,133,134,135,136,137,138]
miR-152	DNMT1/GSTP/CDH1/KIT	CaMKIIα/KIT	[134,135,139,140,141]
miR-15a/16	CCND1/BCL-2/CDK4/6	BCL-2/ARE/CCND1NGN3	[24,142,143,144,145,146,147]
miR-15b	FUT2/GloboH/HNFα	ARE	[146,148,149]
mIR-16	Cyclin D1, NCOR2	ARE/TNFα	[24,142,146,150]
miR-18a	ERα/CTGF	PIAS3	[151,152,153]
miR-192	IL-17/SLC39A6/SNAIL	IL-17RA	[24,154,155,156]
miR-193b	ING5/CCND1/ETS1	TGF-β2	[157,158,159]
miR-200	ZEB1/2		[24,160]
miR-205	ACSL4/E2F1/ZEB1/2		[24,161,162]
miR-21	PTEN/PIP3/AKT/MASPIN/RECK	MYD88/IRAK1/IL-12/SMAD7/PTEN/PDCD4/TPM1	[127,163,164,165,166]
miR-216b	IGF2BP2/IGF2/AKT/mTOR/MAPK/ERK	JAK2	[127,167,168]
miR-222	p27	p27 ^Kip^/KIT	[65,127,145,169]
miR-23a	MYC/CDH1/Sprouty2	IL-4/GATA/FAS	[24,170,171,172,173,174]
miR-26a/c	IL-6/IFNα/ERα/Cyclin D2/Cyclin E2/c-JUN/CDK4/6	IFN-b CDK4/6/MALT1	[63,175,176,177,178]
miR-29c	BCL-2/MCL-1/TNFA1P3	TCL-1/MCL-1/IFN-γ	[179,180,181,182,183]
miR-338-3p	CCND1	ICAM-1	[184,185,186]
miR-34a	CCL22/MAP4K4/SIRT1/CCND1/CDK4/6/MET/C-JUN/CDK2	IFN-b/FOXP1/CDK2/4/6/SIRTI/CCL22/FOXPN	[63,65,187,188,189,190,191,192,193]
miR-363-3p	SPI-1	NO IDENTIFIED IMMUNE TARGET	[24,194]
miR-373	CDH1	MTOR/SIRT1/RELA	[195,196,197,198]
miR-375	AEG-1	JAK2/STAT3	[118]
miR-429	Rab18, NOTCH1	SOX2/BCL-2/SP-1	[199,200,201,202]
miR-520b	HBXIP	RELA	[198,203]
miR-548p	HBXIP, IFN-λ1	IFN-λ1	[204,205]
miR-661	MTA1/NF-κB/iNOS	NO IDENTIFIED IMMUNE TARGET	[23]

**Table 2 viruses-12-00746-t002:** HBx-upregulated miRNA targets in HBV-HCC and immune pathways.

miR	HBV-HCC Target (Hepatocytes)	Immune Target (Hepatocytes/Leucocytes)	Reference
miR-1	MASPIN/HDAC4/E2F5	HDAC4//E2F5/HSP60/HSP70/KCNJ2/GJAJ	[74,206]
miR-107	AXIN2/MASPIN	CDK6	[206,207,208,209]
miR-125a	ERBB2, HBsAg	TNF-α/BCL-2/KLF13/BMF	[9,108,109,140,210,211]
miR-143	FNDC3B	MAPK7	[24,212,213]
miR-146a	CFH/STAT1	IRAK1/TRAF6/IL-1/IRAK2/IL-4/IFN-γ/TIRAP/ NF-κB/IFNγ/STAT1	[6,9,65,126,130,145,214,215,216,217,218,219,220,221,222,223]
miR-155	PTEN/SOX6/ZHX2/SOCS1	IFNγ/SHIP1/SOCSI/BMAL1/PU.1/BACH1/CSFIR/CEBPβ/AID/ETS1	[6,9,145,221,224,225,226,227,228,229,230,231,232,233,234,235,236,237,238]
miR-17-92 family	E2F1, Cyclin G1/PTEN/p21/p27/p57	Th2 induction3/SOCS1/C/EBP/AID/FOXP3/ TNFSF9/CCL-5/IKBKE/c-MAF/AMLI/TP53^INPI^ c-MAF/IFNγ/CD69/PTEN/TGFBR11/p27/p21/E2F/PHLPP2/BIM/CREB1	[6,9,65,145,239,240,241,242,243,244,245]
miR-181a	FAS, E2F5	AID/DUSP5/NLK/PTPN22/SHP2 /DUSP6/CD69/BCL-2	[6,244,246,247,248,249,250,251]
miR-199a-5p	CHC	CD19+	[9,121]
miR-203a	RAP1A	SMAD1/BCL11B/RARB/PRKCA/PRKCB1/FMRP/	[252]
miR-21	PDCD4, PTEN, MASPIN, RECK	PTEN/MYD88/IRAK1/PDCD4/SMAD7	[65,127,164,206,253,254,255,256,257]
miR-215	PTPRP	NO IDENTIFIED IMMUNE TARGET	[154,216,258]
miR-221	ERα, DDIT4/BMF/p27/p57/PTEN/p21/SOCS3	PTEN/SOCS3/p57/KIT/p27 ^kip-^	[65,102,145,169,259,260,261,262,263,264]
miR-222	P27^kip−1/^PTEN/PPP2R2A/p57/p21	p27 ^kip−1^/PTEN/KIT	[127,145,169,261,264,265,266,267]
miR-224	PAK4/MMP9 inhibitor-5/SMAD4	AP15/SMAD4	[268,269,270,271]
miR-27a	PPARγ/FOXO1/APC/P53/RXRα	IL-4/PPARγ	[272,273,274,275]
miR-29a/b	PTEN/PI3K/AKT/MMP-2	IFNARI/ IFN/T-Bet/EOMES/PTEN/MCL-1/ IFN-γ/SLFN4/DNMT3/CDC42/HBP1/TCL1	[127,145,182,183,276,277,278]
miR-30c	HMBOX1	PRDMI/P53	[60,279,280,281]
miR-331-3p	ING5	E2F1/	[282,283]
miR-545/374a	ESRRG	AKT1	[284,285,286]
miR-602	RASSF1a/STAT3/MYC	NO IDENTIFIED IMMUNE TARGET	[11,287]
miR-7	EGFR/RAF/ERK/PI3K-AKT/MASPIN/MTOR	MTOR/CD98/EGFR/TGB-1	[206,285,288,289,290,291,292]

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
