# Peer review of "The Multiple Roles of Hepatitis B Virus X Protein (HBx) Dysregulated MicroRNA in Hepatitis B Virus-Associated Hepatocellular Carcinoma (HBV-HCC) and Immune Pathways"

_viruses, 2020, doi:10.3390/v12070746_

Round 1

Reviewer 1 Report

This review compiles lists of miRNAs that are de-regulated by HBx protein in HBV-associated hepatocellular carcinoma (HCC) and which have roles in modulating gene expression in cells of the immune system. The purpose is to relate pathways affected by HBx expression and pathways of the innate and adaptive response with the development of HCC. This is an ambitious undertaking given the complexities of the networked gene regulatory pathways and cellular interactions that together guide the activities of HBx, miRNAs, immune responses, and HCC development in HBV infected individuals. This review hopes to provide a basis for future studies to unravel these complexities.

The authors appear to be trying to make connections between genes regulated by miRNAs modulated by HBx in hepatocytes and effects on signaling pathways in HCC as well as in cells of the immune system that can be modulated by those same miRNAs. This argument is clouded by a lack of justification for any connection between miRNAs modulated by HBx in hepatocytes and regulation of the expression of those miRNAs in other uninfected cells in the tissue microenvironment, including cells of the immune system. Why have HBx-modulated miRNAs been selected for comment with regard to their role in uninfected immune cells?

The column headings in the tables are confusing. What cell populations are being referenced with the heading "target"? Presumably the HBV-HCC targets are in hepatocytes. Are the immune targets in cells of the immune system or do they refer to hepatocyte pathways that can trigger an immune response? What do the authors conclude from the data in the tables beyond the well known fact that miRNAs can have many targets?

While the task of compiling the information presented in the tables, and summarized in the figures, is already a large one, it would be more informative if it were curated in some way. For instance, the authors might consider the degree of purported modulation of miRNA expression as well as the relative abundance of the miRNAs in the relevant cell types. Are the changes in expression large enough to support changes in the function of pathways where these miRNAs might play a role?

The thumbnail sketches that review the the role of miRNAs in immune cell subpopulations may be useful, but it is not clear how they relate to the title of the review which promises to relate HBx modulation of miRNAs (in infected hepatocytes) to immune pathways (in uninfected immune cells).

The immune response to HBV infection plays a large but incompletely understood role in the progression of liver disease, including the development of HBV-HCC. Changes in miRNA expression in immune cells is a significant element in the immune response as disease progresses, and so considering these changes in the context of advancing liver disease is important. What is missing in this review is the connection, promised in the title, between HBx expression and the immune cell miRNAs selected for discussion. In the absence of that connection, this appears to be a disconnected series of statements about a subset of miRNAs and their potential role in altering the activity of subpopulations of immune cells. The authors add some speculation about biomarkers that further detracts from the focus of this review.

A large amount of data has been collected in this review. It needs a more clear statement of the logic of the connections that the authors are trying to make in order to maximize its value to readers.

Author Response

Reviewer 1

General Comment This review compiles lists of miRNAs that are de-regulated by HBx protein in HBV-associated hepatocellular carcinoma (HCC) and which have roles in modulating gene expression in cells of the immune system. The purpose is to relate pathways affected by HBx expression and pathways of the innate and adaptive response with the development of HCC. This is an ambitious undertaking given the complexities of the networked gene regulatory pathways and cellular interactions that together guide the activities of HBx, miRNAs, immune responses, and HCC development in HBV infected individuals. This review hopes to provide a basis for future studies to unravel these complexities. The authors appear to be trying to make connections between genes regulated by miRNAs modulated by HBx in hepatocytes and effects on signaling pathways in HCC as well as in cells of the immune system that can be modulated by those same miRNAs. This argument is clouded by a lack of justification for any connection between miRNAs modulated by HBx in hepatocytes and regulation of the expression of those miRNAs in other uninfected cells in the tissue microenvironment, including cells of the immune system.

General response: Dear reviewer you prompted us to have a good rethink, sometimes it is difficult to see the wood from the trees. In fact, the comments go to the core of whether the manuscript would be useful or not to future readers. Thank you, we have tried to make amends as per following responses and revisions to the text and have stressed the hypothetical element of the paper needs further studies . For example re your second paragraph above, in the introduction we nail our colours to the mast as follows”. The principal focus of this exploratory review is to examine the complex role of some key miRNA that are dysregulated by the HBx  protein of HBV in the HBV-HCC continuum, as well as in both the innate and adaptive immune cells. In this regard, our focus is to demonstrate how the HBx protein can dysregulate miRNA in hepatocytes in HBV-HCC pathogenesis and how this can simultaneously trigger changes in the same miRNA expression in innate and adaptive immune cell pathways. This is the connection we seek to make, namely, that in HBV-HCC pathogenesis the miRNA response in immune cells is not independent of their expression in hepatocytes. We, therefore, hypothesize that in HBV-HCC pathogenesis specific HBx dysregulated miRNA in hepatocytes also become dysregulated in immune cells because of the influence of viral infection. This study provides a platform for multiple hypotheses for future studies.

In section 6 we try to link section 5-7 and state” This section demonstrates the regulatory role of specific miRNA in specific innate and adaptive cell pathways and contrasts with Section 5 which illustrated the target genes of HBx dysregulated miRNA in hepatocytes in HBV-HCC pathogenesis, as well as some of their validated immune targets in both hepatocytes and immune cells”  and at the end of this section we say” In the next section, we demonstrate that the same HBX dysregulated miRNA in HBV-HCC in hepatocytes can be interdependently activated in the innate and adaptive cell pathways.  

We then provide an expanded conclusion as per our response to your question 5 below

Reviewer Question 1: Why have HBx-modulated miRNAs been selected for comment with regard to their role in uninfected immune cells?

Response: We have clarified an important point that you raise and hopefully better stated our position in the resulting revision. The presence of HBx/HBV /inflammation/damage infected hepatocytes elicits a miRNA response in immune cells. Typically, immune cells respond to the presence of a pathogen e.g. “Through recognition of viral nucleic acids, viral proteins or tissue-damage innate immunity is triggered during the early phases of viral infections. Activation of different families of cellular receptors (toll-like receptors [TLRs], RIG-1) leads to rapid production of antiviral cytokines, such as interferon (IFN)-α, and, in concert with activation of natural killer (NK) cells, limits the initial spread of hepatitis B virus (HBV)”. In turn, miRNA expression in these immune cells can be dysregulated for a range of reasons, e.g. to modulate pro and anti-inflammatory response etc. This is the connection we seek to make, namely, that the miRNA response in immune cells is NOT independent of HBV-HCC pathogenesis in hepatocytes.

Reviewer Question 2: The column headings in the tables are confusing. What cell populations are being referenced with the heading "target"? Presumably the HBV-HCC targets are in hepatocytes. Are the immune targets in cells of the immune system or do they refer to hepatocyte pathways that can trigger an immune response?

Response:  The cell populations in the second column refer to both immune gene targets in hepatocytes and immune cells that are targeted by a specific miRNA. In the specific examples that are expanded in the figures the miRNA targets in the immune system are based on studies of innate and immune cell response in cancer in general and HCC. This is now clarified in the revised manuscript. 

Reviewer Question 3: What do the authors conclude from the data in the tables beyond the well known fact that miRNAs can have many targets?

Answer: We really wanted in a general way to emphasize that nearly all HBx dysregulated miRNA in HBV-HCC can potentially target multiple immune targets and now emphasize this in the conclusion in order to justify the use of the tables. The second column in T1&2, therefore,  illustrates potential immune targets in both hepatocytes and leukocytes and underlines the multiple regulatory roles of the same miRNA and targets in different cell types including those in the immune system 

Reviewer Question 4: While the task of compiling the information presented in the tables, and summarized in the figures, is already a large one, it would be more informative if it were curated in some way. For instance, the authors might consider the degree of purported modulation of miRNA expression as well as the relative abundance of the miRNAs in the relevant cell types. Are the changes in expression large enough to support changes in the function of pathways where these miRNAs might play a role?

Response: This key question is what is really holding back the exploitation of miRNA as therapeutic agents. Why the miRNA field remains a work-in-progress. For starters, miRNA act as a mild ancillary immune system, therefore, the probability is that constellations of miRNA collectively manage modulation rather than individual candidates. For instance if one looks at our candidate miRNA across all cancers, miR-155 is on average upregulated 0.36 fold, miR-17-92 family from 0.35 to 0.85, miR-181 at 0.03, miR-21 at 0.84. Most of the cell line studies, therefore show their impact at various dosages which are not corroborated by in vivo studies.  Please see highlighted section of conclusion that questions exactly what you are saying. 

Reviewer Question 5: The thumbnail sketches that review the role of miRNAs in immune cell subpopulations may be useful, but it is not clear how they relate to the title of the review which promises to relate HBx modulation of miRNAs (in infected hepatocytes) to immune pathways (in uninfected immune cells).

Response: I am hoping that our response to your question (general comment + Reviewer Question 1) will suffice together with the changes we have made to both the text and the title. In addition, we now spell out in each section from section 5 to 7 what we are trying to do and conclude the following in the conclusion “In this extensive review we have attempted to bring together studies, which have shown  the complex interlinking roles of miRNA in HBV-HCC pathogenesis and the immune response, both innate and adaptive. Moreover, from the literature it is evident that nearly all HBx dysregulated miRNA in HBV-HCC can additionally act on multiple immune targets (Tables 1& 2). Using four key miRNA as an illustration it is clear that there is simultaneous modulation of central pathways, namely, the principal HBV-HCC cancer pathways and those of the innate and adaptive immune systems (Figures 1-4). We, therefore, hypothesize that the same specific miRNA that are dysregulated in hepatocytes during HBV-HCC pathogenesis can become simultaneously and interdependently dysregulated in immune cells and vice versa. The four representative miRNA selected primarily demonstrate how they modulate HBV replication and oncogene or tumour suppressor expression in HBV-HCC pathogenesis while simultaneously modulating the proliferation and differentiation of leucocytes in the innate and adaptive immune systems. This interplay between the two pathways may provide us with the possibility of using candidate miRNA to manipulate this interaction as a potential therapeutic option.

Multiple miRNA target the same genes and post-transcriptional gene silencing of translation is a collective effort. Even then it is likely miRNA only exert a mild secondary influence on mRNA stability and translation in response to the stochastic nature of gene expression and changing environmental influences (7). Furthermore, small tumors (< 0.5cm) would be unable by themselves to alter the level of extracellular miRNA in sera and the explanation for dysregulated miRNA in early stage carcinogenesis would likely be as a result of general immune responses (375). In vivo results also indicate that most RNA-based therapies are compromised by non-specific organ bio-distribution, reticuloendothelial system (RES) clearance, and endolysosomal trafficking (376). Increasingly, future studies will need to consider the selection of sub-populations of extracellular vesicles that facilitate small RNA messaging. Emerging research indicates that only certain types of encapsulated miRNA play a role in cell-cell signaling and others may not. Exosomes, for instance, appear to transport miRNA that promote paracrine communication (377-379) and nanotechnology can be used to deliver chemically modified miRNA to cancer cells (380, 381).

This rather simplistic account cannot illustrate the full extent of the dynamic, complex and multi-dimensional role of each miRNA in varying HBV-HCC cases either with respect to the varying degrees of expression in each pathway or the degree to which HBV-HCC pathogenesis can be modulated. However, the demonstration of these interrelationships will allow each of these potential interactions to be treated as hypotheses that need to be tested individually.  Although miRNA hold a promise as therapeutic agents in various cancers including HBV-associated HCC, this field of study remains a work in progress that is yet to be fully exploited (382).

 Reviewer Comment: The immune response to HBV infection plays a large but incompletely understood role in the progression of liver disease, including the development of HBV-HCC. Changes in miRNA expression in immune cells is a significant element in the immune response as disease progresses, and so considering these changes in the context of advancing liver disease is important. What is missing in this review is the connection, promised in the title, between HBx expression and the immune cell miRNAs selected for discussion. In the absence of that connection, this appears to be a disconnected series of statements about a subset of miRNAs and their potential role in altering the activity of subpopulations of immune cells. The authors add some speculation about biomarkers that further detracts from the focus of this review.

Response: In the revised manuscript, leading on from your general comment and question 1 which seems to be the central issue that concerns you, we have attempted to make this link clearer (as per our responses to above questions. We have stressed, however, that the candidate miRNA in specific innate and adaptive pathways in HBV-HCC need to be individually tested and validated and hope that this manuscript prompts numerous hypotheses for future studies. We have deleted reference to miRNA as diagnostic agents

Reviewer Comment: A large amount of data has been collected in this review. It needs a more clear statement of the logic of the connections that the authors are trying to make in order to maximize its value to readers.

Response: As per above

Reviewer 2 Report

This review article entitled “The multiple roles of Hepatitis B Virus X Protein (HBx) dysregulated microRNA in Hepatitis B Virus-associated Hepatocellular carcinoma (HBV-HCC) and immune pathways” by Kurt et al. ambitiously reviewed 382 papers focusing on those microRNA which dysregulated by HBx during HBV induced HCC. Especially they focus on immune response related microRNA. There is a similar review paper "The Regulatory Role of MicroRNA in Hepatitis-B Virus-Associated Hepatocellular Carcinoma (HBV-HCC) Pathogenesis” published by the same group on Cells. 2019 Nov 24;8(12):1504. I found those two review papers are quite similar, there are many parts in the entire article is highly similar to those published papers.

The subtitle 1(Background) first two lines and last four lines (miRNA) act as post-transcriptional gene silencers that collectively reduce or inhibit their target mRNA expression, thereby playing a homeostatic role that fine tunes the translation,  proteins. The ancillary role of miRNA, as mild suppressors, has been explained by the inherently stochastic nature of gene transcription and environmental fluctuations) are exactly similar to 2019 Cells paper.

Subtitle 2 (HBV-HCC pathogenesis and miRNA expression) the first paragraph is almost exactly the same as the 2019 Cells paper, last paragraph also is very similar to 2019 Cells.

Subtitle 5, the last four lines of the first paragraph is copied from 2019 Cells paper. HBV-HCC pathways typically include aberrant expression in the retinoblastoma-tumor protein 53 (RB1-TP53) suppressor networks, the Wingless-related integration site/beta-catenin (WNT/β-Catenin) pathway, and the phosphoinositide 3-kinase/mitogen-activated protein kinase (PI3K/MAPK) and Janus kinase/signal transducer (JAK/STAT) pathways

Subtitle 6, the first two lines also showed 100% similarity.

In the case of transient reaction to environmental conditions, miRNA quantity become temporarily dysregulated until homeostasis is restored.

Subtitle 7.1, last two lines of the second paragraph is same as 2019 Cells, “ HBx upregulated miR-155 also subdues HBV replication by blocking the CCAAT/enhancer-binding protein (C/EBPthat binds and activates the HBV Enhancer (Enh) 11/core promoter”, the middle part of the the paragraph before subtitle 8, author contribution, funding are also 100% similar to their 2019 Cells paper. The authors should try to rephrase those sentences.

Subtitle 7.1, miR-155 has distinct expression profiles and plays a crucial role in various physiological and pathological processes such as haematopoietic lineage differentiation, immunity, inflammationcancer

In the subtitle “B-cell”, the first three lines of the first paragraph is exactly the same as “The control of B cell development in the bone marrow depends on the commitment of progenitor cells to the B cell lineage by the activation of transcription factor networks, V(D)J recombination events and selection for effective antigen receptors.” which is published on Oncogene volume 34, pages3085–3094(2015). Second paragraph third to fifth lines also is exactly the same as “miR-155-deficient B cells have defective antibody class switching and differentiation into plasma cells, resulting in an impaired humoral response to T cell-dependent antigenic stimulation” which is published on Oncogene volume 34, pages3085–3094(2015).

Subtitle 7.2, second paragraph, is the same as ”E2F3 increases the expression of miR-17–92 miRNAs, which in turn negatively regulates the translation of E2F3 and E2F1. This may represent a mechanism that enables transformed hepatocytes to circumvent the apoptotic effects of E2F1 accumulation” which was published on American Journal of Pathology.

Subtitle 4, first paragraph, highly similar to Oncogene volume 34, pages3085–3094(2015) “Various mechanisms contribute to aberrant miRNA expression in cancer.13,14 As with abnormalities in oncogenes and tumor-suppressor genes, alterations in miRNAs can be explained in part by several mechanisms, including chromosomal deletion, amplification, mutation, epigenetic silencing and transcriptional dysregulation of pri-miRNA transcripts.”

modulate the microenvironment via non-cell-autonomous mechanisms, and alterations in the miRNA profiles of neighboring cells”

” CAFs, in contrast to normal fibroblasts, enhance ECM production and secrete cancer-activating cytokines and chemokines, significantly promoting tumorigenesis

Regarding the presentation, it is difficult to fully understand and digest the content in the current presentation. Some sentence are too long and need to be more concise. For example, the second sentence of the subtitle 5, it covered four lines.

The resolution of the images should be improved, so the readers can fully appreciate the networks.

Author Response

Reviewer General Comment: This review article entitled “The multiple roles of Hepatitis B Virus X Protein (HBx) dysregulated microRNA in Hepatitis B Virus-associated Hepatocellular carcinoma (HBV-HCC) and immune pathways” by Kurt et al. ambitiously reviewed 382 papers focusing on those microRNA which dysregulated by HBx during HBV induced HCC. Especially they focus on immune response related microRNA. There is a similar review paper "The Regulatory Role of MicroRNA in Hepatitis-B Virus-Associated Hepatocellular Carcinoma (HBV-HCC) Pathogenesis” published by the same group on Cells. 2019 Nov 24;8(12):1504. I found those two review papers are quite similar, there are many parts in the entire article is highly similar to those published papers.

Response General Comment: Yes you are right, the use of exactly the same text from the Cells 2019 article is sloppy and can appear to be “salami slicing” in order to get two publications for the price of one. In fact, the regulatory role of miRNA in the Cells article provided the comparative background of their role in HBV-HCC which we wanted to compare with their role in immune cells. In the revisions you suggest we have ensured that the HBV-HCC background is completely original.  We have also thoroughly revised the manuscript to better explain the central purpose of the study (see our response to your final comment). Our Biorender figures have been upgraded to high quality in final version.

Comment 1: The subtitle 1(Background) first two lines and last four lines (miRNA) act as post-transcriptional gene silencers that collectively reduce or inhibit their target mRNA expression, thereby playing a homeostatic role that fine tunes the translation,  proteins. The ancillary role of miRNA, as mild suppressors, has been explained by the inherently stochastic nature of gene transcription and environmental fluctuations) are exactly similar to 2019 Cells paper.

 Response: reworded

Comment 2:Subtitle 2 (HBV-HCC pathogenesis and miRNA expression) the first paragraph is almost exactly the same as the 2019 Cells paper, last paragraph also is very similar to 2019 Cells.

Response: reworded

Comment 3: Subtitle 5, the last four lines of the first paragraph is copied from 2019 Cells paper. HBV-HCC pathways typically include aberrant expression in the retinoblastoma-tumor protein 53 (RB1-TP53) suppressor networks, the Wingless-related integration site/beta-catenin (WNT/β-Catenin) pathway, and the phosphoinositide 3-kinase/mitogen-activated protein kinase (PI3K/MAPK) and Janus kinase/signal transducer (JAK/STAT) pathways

Response: reworded

Comment 4: Subtitle 6, the first two lines also showed 100% similarity.

In the case of transient reaction to environmental conditions, miRNA quantity become temporarily dysregulated until homeostasis is restored.

Response: whole paragraph re-worded

Comment 5; Subtitle 7.1, last two lines of the second paragraph is same as 2019 Cells, “ HBx upregulated miR-155 also subdues HBV replication by blocking the CCAAT/enhancer-binding protein (C/EBPthat binds and activates the HBV Enhancer (Enh) 11/core promoter”, the middle part of the the paragraph before subtitle 8, author contribution, funding are also 100% similar to their 2019 Cells paper. The authors should try to rephrase those sentences.

Response: reworded

Comment 6:Subtitle 7.1, miR-155 has distinct expression profiles and plays a crucial role in various physiological and pathological processes such as haematopoietic lineage differentiation, immunity, inflammationcancer

 Response: reworded

Comment 7: In the subtitle “B-cell”, the first three lines of the first paragraph is exactly the same as “The control of B cell development in the bone marrow depends on the commitment of progenitor cells to the B cell lineage by the activation of transcription factor networks, V(D)J recombination events and selection for effective antigen receptors.” which is published on Oncogene volume 34, pages3085–3094(2015).

Response: paragraph changed

Comment 8: Second paragraph third to fifth lines also is exactly the same as “miR-155-deficient B cells have defective antibody class switching and differentiation into plasma cells, resulting in an impaired humoral response to T cell-dependent antigenic stimulation” which is published on Oncogene volume 34, pages3085–3094(2015).

Response: reworded 

Comment 9: Subtitle 7.2, second paragraph, is the same as ”E2F3 increases the expression of miR-17–92 miRNAs, which in turn negatively regulates the translation of E2F3 and E2F1. This may represent a mechanism that enables transformed hepatocytes to circumvent the apoptotic effects of E2F1 accumulation” which was published on American Journal of Pathology.

 Response: reworded

Comment 10: Subtitle 4, first paragraph, highly similar to Oncogene volume 34, pages3085–3094(2015) “Various mechanisms contribute to aberrant miRNA expression in cancer.13,14 As with abnormalities in oncogenes and tumor-suppressor genes, alterations in miRNAs can be explained in part by several mechanisms, including chromosomal deletion, amplification, mutation, epigenetic silencing and transcriptional dysregulation of pri-miRNA transcripts.”

Response: reworded

Comment 11: ” modulate the microenvironment via non-cell-autonomous mechanisms, and alterations in the miRNA profiles of neighboring cells”

” CAFs, in contrast to normal fibroblasts, enhance ECM production and secrete cancer-activating cytokines and chemokines, significantly promoting tumorigenesis

 Response: reworded

Comment 12: Regarding the presentation, it is difficult to fully understand and digest the content in the current presentation. Some sentence are too long and need to be more concise. For example, the second sentence of the subtitle 5, it covered four lines.

Response: We have substantially revised the manuscript and hope that the central point of this review is now clearer, namely in introduction, “The principal focus of this exploratory review is to examine the complex role of some key miRNA that are dysregulated by the HBx  protein of HBV in the HBV-HCC continuum, as well as in both the innate and adaptive immune cells. In this regard, our focus is to demonstrate how the HBx protein can dysregulate miRNA in hepatocytes in HBV-HCC pathogenesis and how this can simultaneously trigger changes in the same miRNA expression in innate and adaptive immune cell pathways. This is the connection we seek to make, namely, that in HBV-HCC pathogenesis the miRNA response in immune cells is not independent of their expression in hepatocytes. We, therefore, hypothesize that in HBV-HCC pathogenesis specific HBx dysregulated miRNA in hepatocytes also become dysregulated in immune cells because of the influence of viral infection. This study provides a platform for multiple hypotheses for future studies.

 in Section 5 “ In HBV-HCC, the tumour micro-environment dysregulates a range of miRNA in hepatocytes to modulate pathogenesis. Simultaneously, innate and adaptive immune cells respond to the presence of the tumour micro-environment. This response across different cell types occurs via the recognition of viral nucleic acids, viral proteins or tissue-damage and results in the activation of different families of cellular receptors (34). In turn, the same miRNA that can be dysregulated in hepatocytes are also expressed in these immune cells and can be dysregulated for a range of reasons, e.g. to modulate pro and anti-inflammatory response (58).  Moreover, we now link what we are trying to achieve in sections 5-7

In conclusion “In this extensive review we have attempted to bring together studies, which have shown  the complex interlinking roles of miRNA in HBV-HCC pathogenesis and the immune response, both innate and adaptive. Moreover, from the literature it is evident that nearly all HBx dysregulated miRNA in HBV-HCC can additionally act on multiple immune targets (Tables 1& 2). Using four key miRNA as an illustration it is clear that there is simultaneous modulation of central pathways, namely, the principal HBV-HCC cancer pathways and those of the innate and adaptive immune systems (Figures 1-4). We, therefore, hypothesize that the same specific miRNA that are dysregulated in hepatocytes during HBV-HCC pathogenesis can become simultaneously and interdependently dysregulated in immune cells and vice versa. The four representative miRNA selected primarily demonstrate how they modulate HBV replication and oncogene or tumour suppressor expression in HBV-HCC pathogenesis while simultaneously modulating the proliferation and differentiation of leucocytes in the innate and adaptive immune systems. This interplay between the two pathways may provide us with the possibility of using candidate miRNA to manipulate this interaction as a potential therapeutic option.

Multiple miRNA target the same genes and post-transcriptional gene silencing of translation is a collective effort. Even then it is likely miRNA only exert a mild secondary influence on mRNA stability and translation in response to the stochastic nature of gene expression and changing environmental influences (7). Furthermore, small tumors (< 0.5cm) would be unable by themselves to alter the level of extracellular miRNA in sera and the explanation for dysregulated miRNA in early stage carcinogenesis would likely be as a result of general immune responses (375). In vivo results also indicate that most RNA-based therapies are compromised by non-specific organ bio-distribution, reticuloendothelial system (RES) clearance, and endolysosomal trafficking (376). Increasingly, future studies will need to consider the selection of sub-populations of extracellular vesicles that facilitate small RNA messaging. Emerging research indicates that only certain types of encapsulated miRNA play a role in cell-cell signaling and others may not. Exosomes, for instance, appear to transport miRNA that promote paracrine communication (377-379) and nanotechnology can be used to deliver chemically modified miRNA to cancer cells (380, 381).

This rather simplistic account cannot illustrate the full extent of the dynamic, complex and multi-dimensional role of each miRNA in varying HBV-HCC cases either with respect to the varying degrees of expression in each pathway or the degree to which HBV-HCC pathogenesis can be modulated. However, the demonstration of these interrelationships will allow each of these potential interactions to be treated as hypotheses that need to be tested individually.  Although miRNA hold a promise as therapeutic agents in various cancers including HBV-associated HCC, this field of study remains a work in progress that is yet to be fully exploited (382).

 Comment 13: The resolution of the images should be improved, so the readers can fully appreciate the networks.

Response: we have upgraded Biorender figures to high quality.

Reviewer 3 Report

June 24, 2020

Review for VIRUSES.

 “The multiple roles of Hepatitis B virus x protein (HBx) dysregulated microRNA in Hepatitis B-associated hepatocellular carcinoma (HBV-HCC) and immune pathways.

General comment

Authors have conducted an excellent and comprehensive review on the HBx dysregulated miRNAs and their potential roles in the pathogenesis of in HBV associated HCC and examined the immune pathways.

The authors are highly commended for their massive work including the extensive review of nearly 400 references.  They provided the detailed description of the known miRNAs involved in HBV-HCC.

While their detailed review provided probable immune pathways, the presented immune pathways was treated as hypotheses. Therefore, the authors encourage further investigations in the future. 

It is hoped that with further investigation and in depth understanding of the role of MiRs in HBV associated hepatocarinogenesis, one should be able to prevent HBV associated carcinogenesis ( in addition to HBV vaccination) but also the effective treatment for the already developed HBV-HCC.

Minor Question:

When miRNA and immune system was described in detail, the immune system was divided into innate and adaptive, and discussed. 

When well-characterized miRNAs (miR-155, miR-17-92, miR-181a and miR-21) were described individually, only with miR-155, the separation of two systems was continued. For the rest of miRs, all were placed under the Innate immune system including T and B cells which, I believe, should be under the heading of adaptive immune system. It needs to be clarified. 

Author Response

Authors have conducted an excellent and comprehensive review on the HBx dysregulated miRNAs and their potential roles in the pathogenesis of in HBV associated HCC and examined the immune pathways.

The authors are highly commended for their massive work including the extensive review of nearly 400 references.  They provided the detailed description of the known miRNAs involved in HBV-HCC.

While their detailed review provided probable immune pathways, the presented immune pathways was treated as hypotheses. Therefore, the authors encourage further investigations in the future. 

It is hoped that with further investigation and in depth understanding of the role of MiRs in HBV associated hepatocarinogenesis, one should be able to prevent HBV associated carcinogenesis ( in addition to HBV vaccination) but also the effective treatment for the already developed HBV-HCC.

Response: many thanks for your review. We have made quite a few changes to the manuscript, in particular, tried to better explain the exact purpose of the study. These changes have been highlighted in the revised text and the key point we wanted to emphasize was “This is the connection we seek to make, namely, that the miRNA response in immune cells is not independent of HBV-HCC pathogenesis in hepatocytes”   

Minor Question:

When miRNA and immune system was described in detail, the immune system was divided into innate and adaptive, and discussed. 

When well-characterized miRNAs (miR-155, miR-17-92, miR-181a and miR-21) were described individually, only with miR-155, the separation of two systems was continued. For the rest of miRs, all were placed under the Innate immune system including T and B cells which, I believe, should be under the heading of adaptive immune system. It needs to be clarified. 

Response: this has now been corrected for each of the four selected miRNA

Reviewer 4 Report

In this review, Sartorius et al. summarized HBx dysregulated miRNAs and the roles of these miRNAs in HBV-associated HCC and immune pathways. Tremendous literature has been summarized in this article. However, this article lacks authors’ commentaries or opinions on the relations between miRNAs in immune pathways and these miRNAs in HBV-HCC pathogenesis, which would make this article more interesting to readers. Thus, this paper is more like a literature summary instead of a review. I would suggest the authors add some of the opinions or comments on each part. This manuscript would benefit from the more careful editing of grammar.

Comments:

  1. There are many abbreviations used in the context, please double check you have used the same form. For example, NFkB should be NF-κB, beta-catenin should consistently use β-Catenin or beta-catenin, etc.
  2. Use the full name when firstly use the abbreviation, such as GC for the germinal center, TFH for the T-follicular helper cells, etc. Please go through the text and check all abbreviations when first use them.
  3. Use miRNAs instead of miRNA in some sentences.
  4. Other comments see the PDF file.

Author Response

Reviewer 4

Reviewer comment: In this review, Sartorius et al. summarized HBx dysregulated miRNAs and the roles of these miRNAs in HBV-associated HCC and immune pathways. Tremendous literature has been summarized in this article. However, this article lacks authors’ commentaries or opinions on the relations between miRNAs in immune pathways and these miRNAs in HBV-HCC pathogenesis, which would make this article more interesting to readers. Thus, this paper is more like a literature summary instead of a review. I would suggest the authors add some of the opinions or comments on each part. This manuscript would benefit from the more careful editing of grammar.

Response: This we also felt was a central problem that is emphasized in a different way by Reviewer 1, namely, that the true purpose of the study was not clear, why did we pick HBx dysregulated miRNA, what cell types are we talking about. What can be proposed by the authors to change this literature summary into a review. Please see extensive changes made to manuscript e.g. In response we have changed as follows:

in introduction, “The principal focus of this exploratory review is to examine the complex role of some key miRNA that are dysregulated by the HBx  protein of HBV in the HBV-HCC continuum, as well as in both the innate and adaptive immune cells. In this regard, our focus is to demonstrate how the HBx protein can dysregulate miRNA in hepatocytes in HBV-HCC pathogenesis and how this can simultaneously trigger changes in the same miRNA expression in innate and adaptive immune cell pathways. This is the connection we seek to make, namely, that in HBV-HCC pathogenesis the miRNA response in immune cells is not independent of their expression in hepatocytes. We, therefore, hypothesize that in HBV-HCC pathogenesis specific HBx dysregulated miRNA in hepatocytes also become dysregulated in immune cells because of the influence of viral infection. This study provides a platform for multiple hypotheses for future studies.  

in Section 5 “ In HBV-HCC, the tumour micro-environment dysregulates a range of miRNA in hepatocytes to modulate pathogenesis. Simultaneously, innate and adaptive immune cells respond to the presence of the tumour micro-environment. This response across different cell types occurs via the recognition of viral nucleic acids, viral proteins or tissue-damage and results in the activation of different families of cellular receptors (34). In turn, the same miRNA that can be dysregulated in hepatocytes are also expressed in these immune cells and can be dysregulated for a range of reasons, e.g. to modulate pro and anti-inflammatory response (58).  We now also explain link sections 5 to 7

In conclusion” In this extensive review we have attempted to bring together studies, which have shown  the complex interlinking roles of miRNA in HBV-HCC pathogenesis and the immune response, both innate and adaptive. Moreover, from the literature it is evident that nearly all HBx dysregulated miRNA in HBV-HCC can additionally act on multiple immune targets (Tables 1& 2). Using four key miRNA as an illustration it is clear that there is simultaneous modulation of central pathways, namely, the principal HBV-HCC cancer pathways and those of the innate and adaptive immune systems (Figures 1-4). We, therefore, hypothesize that the same specific miRNA that are dysregulated in hepatocytes during HBV-HCC pathogenesis can become simultaneously and interdependently dysregulated in immune cells and vice versa. The four representative miRNA selected primarily demonstrate how they modulate HBV replication and oncogene or tumour suppressor expression in HBV-HCC pathogenesis while simultaneously modulating the proliferation and differentiation of leucocytes in the innate and adaptive immune systems. This interplay between the two pathways may provide us with the possibility of using candidate miRNA to manipulate this interaction as a potential therapeutic option.

Multiple miRNA target the same genes and post-transcriptional gene silencing of translation is a collective effort. Even then it is likely miRNA only exert a mild secondary influence on mRNA stability and translation in response to the stochastic nature of gene expression and changing environmental influences (7). Furthermore, small tumors (< 0.5cm) would be unable by themselves to alter the level of extracellular miRNA in sera and the explanation for dysregulated miRNA in early stage carcinogenesis would likely be as a result of general immune responses (375). In vivo results also indicate that most RNA-based therapies are compromised by non-specific organ bio-distribution, reticuloendothelial system (RES) clearance, and endolysosomal trafficking (376). Increasingly, future studies will need to consider the selection of sub-populations of extracellular vesicles that facilitate small RNA messaging. Emerging research indicates that only certain types of encapsulated miRNA play a role in cell-cell signaling and others may not. Exosomes, for instance, appear to transport miRNA that promote paracrine communication (377-379) and nanotechnology can be used to deliver chemically modified miRNA to cancer cells (380, 381).

This rather simplistic account cannot illustrate the full extent of the dynamic, complex and multi-dimensional role of each miRNA in varying HBV-HCC cases either with respect to the varying degrees of expression in each pathway or the degree to which HBV-HCC pathogenesis can be modulated. However, the demonstration of these interrelationships will allow each of these potential interactions to be treated as hypotheses that need to be tested individually.  Although miRNA hold a promise as therapeutic agents in various cancers including HBV-associated HCC, this field of study remains a work in progress that is yet to be fully exploited (382).

Please see final revised manuscript. We hope this adds a bit more opinion and that this can promote the value of our review bearing in mind that we say that the hypothesized immune pathways need to be individually tested.

Comments:

  1. There are many abbreviations used in the context, please double check you have used the same form. For example, NFkB should be NF-κB, beta-catenin should consistently use β-Catenin or beta-catenin, etc.
  2. Use the full name when firstly use the abbreviation, such as GC for the germinal center, TFH for the T-follicular helper cells, etc. Please go through the text and check all abbreviations when first use them.

Response: corrected

  1. Use miRNAs instead of miRNA in some sentences.

Response: I have been told by my miRNA people that I must consistently use miRNA or miRNAs throughout the text, that it can be singular or plural

  1. Other comments see the PDF file.

Response: Purpose in Section 1 (introduction) completely re-written, suggested edit adopted and where reviewer unhappy with wording alternate wording use. Authors advised to either use miRNA or miRNAs consistently as miRNA can be singular or plural.

Edits in Section 2 adopted except enzyme left in text.  Edits in Section 3 completed. Edits in section 4 completed. Authors advised to either use miRNA or miRNAs consistently as this can be singular or plural. In section 5 before Table 1 paragraph re-written. In Section 6 deleted GC edits made, ditto Section 7 edits made, zero changed to zone

Round 2

Reviewer 2 Report

This revised manuscript improved a lot, the images are higher resolution. However, there are some parts still need to be reworded to avoid the highly similar to the published paper as mentioned below:

Following are similar to Cells 20198(12), 1504 in bold:

Subtitle 2, first paragraph

a spectrum of clinical manifestations including the asymptomatic carrier state, acute or fulminant hepatitis,

self-limiting, with the virus persisting in 5% and 90% of the cases, respectively

T-cell responses accompanied by secondary inflammatory response [14], and an increase in free radicals, interferon, tumor necrosis factor (TNF), and hepatic injury

results in their eventual replacement with undifferentiated liver stem cells and poorly organized fibrotic tissue

 deletions, cis/trans-activation, translocations, the production of fusion transcripts, aberrant epigenetic changes, and generalized genomic instability [17]. In 

Subtitle 2, last paragraph

HBV infection leading to HCC, multiple miRNA become increasingly permanently dysregulated

as a result of HBV infection, epigenetic changes [22], inflammation [23], fibrosis [24], cirrhosis [16], and the onset of HCCThe increasing level of dysregulation in the HBV-HCC continuum

showed 37 miRNA deregulated in otherwise healthy controls (HC), 77 in

Subtitle 5, second paragraph:

protein 53 (RB1-TP53) suppressor networks, the Wingless-related integration site/beta-catenin (WNT/β-Catenin) pathway, and the phosphoinositide 3-kinase/mitogen-activated protein kinase (PI3K/MAPKand Janus kinase/signal transducer (JAK/STAT)

macrophage:

The loss of function of SOCS-1 is a common feature in HCC and the HBx-mediated upregulation of miR-155 is a contributing factor in HBV-HCC

subtitle 7.5, second paragraph:

blocks p53 stimulated miR-34 expression in hepatocytes leading to the upregulation of macrophage-derived chemokine (CCL22) that stimulates regulatory T-cells (Tregs) that, in turn, block effector T-cells allowing HBV expression to increase 

Following are similar to World J Gastroenterol. Jun 28, 2014; 20(24): 7971-7978 in bold:

Subtitle 2, third paragraph

"(HBx) is known to play a pivotal role in the pathogenesis of viral induced HCC. HBx is a multifunctional protein of 17 kDa which modulates several cellular processes by direct or indirect interaction with a repertoire of host factors"

"several cellular processes such as oxidative stress, DNA repair, signal transduction, transcription, protein degradation, cell cycle progression and apoptosis." 

Adaptive immune system-T-cell

The frequency of Tregs in the HCC tumor microenvironment was significantly higher than that in tumor-surrounding tissue and biopsy specimens from healthy livers

Following are similar to Oncogene volume 34pages3085–3094(2015) in bold:

Subtitle 6.1, NK cells 

are expressed by cells undergoing stress triggered by events such as viral infection or cell transformation. Engagement of NKG2D on NK cells 

Subtitle 6.1, B cell

control of B cell development in the bone marrow depends on the commitment of progenitor cells to the B cell lineage by the activation of transcription factor networks, V(D)J recombination events and selection for effective antigen receptors

miR-155-deficient B cells have defective antibody class switching and differentiation into plasma cells, resulting in an impaired humoral response to T cell-dependent antigenic

T-cells

(TCR), as occurs during activation-induced cell death, is known to depend on the CD95–CD95 ligand pathway

Author Response

ROUND 2 Review:

This revised manuscript improved a lot, the images are higher resolution. However, there are some parts still need to be reworded to avoid the highly similar to the published paper as mentioned below:

Following are similar to Cells 20198(12), 1504 in bold:

Subtitle 2, first paragraph

a spectrum of clinical manifestations including the asymptomatic carrier state, acute or fulminant hepatitis,

Corrected: HBV infection manifests in a range of clinical conditions including the asymptomatic carrier state, inflammation, acute or fulminant hepatitis, chronic hepatitis, and the onset of cirrhosis

self-limiting, with the virus persisting in 5% and 90% of the cases, respectively

Corrected :Acute HBV infection only persists in 5% of adults , unlike in children where 90% of the cases do not resolve

T-cell responses accompanied by secondary inflammatory response [14], and an increase  

Corrected: T-cell responses in the presence of a secondary inflammatory response, as well as increases  in free radicals, interferon, tumor necrosis factor (TNF) and hepatic injury (13)

results in their eventual replacement with undifferentiated liver stem cells and poorly organized fibrotic tissue

Corrected: results in their eventual depletion and their replacement with liver stem cells and less well organized fibrotic tissue

 deletions, cis/trans-activation, translocations, the production of fusion transcripts, aberrant epigenetic changes, and generalized genomic instability [17].

Corrected: Oncogenic disruption leads to genomic instability that can include aberrant epigenetic change, DNA deletions, fusion transcripts cis/trans-activation, and translocations, (5).

Subtitle 2, last paragraph

HBV infection leading to HCC, multiple miRNA become increasingly permanently dysregulated

Corrected: words added.

as a result of HBV infection, epigenetic changes [22], inflammation [23], fibrosis [24], cirrhosis [16], and the onset of HCCThe increasing level of dysregulation in the HBV-HCC continuum

Corrected: multiple miRNA become increasingly permanently dysregulated as a result of HBV infection, inflammation (26), fibrosis (27), cirrhosis (15) and the onset of HCC (28). The increasing C

showed 37 miRNA deregulated in otherwise healthy controls (HC), 77 in

Corrected: while another study demonstrated an increase from 37 miRNA deregulated in healthy controls (HC) to 77

Subtitle 5, second paragraph:

protein 53 (RB1-TP53) suppressor networks, the Wingless-related integration site/beta-catenin (WNT/β-Catenin) pathway, and the phosphoinositide 3-kinase/mitogen-activated protein kinase (PI3K/MAPKand Janus kinase/signal transducer (JAK/STAT)

Corrected: These include the Retinoblastoma-Tumour Protein 53 (RB1-TP53) suppressor networks, the Phosphoinositide 3-kinase / mitogen-activated protein kinase (PI3K/MAPK) pathway, the Wingless related integration site/beta-Catenin (WNT/β-Catenin) pathway and the Janus kinase/signal transducer (JAK/STAT) pathway (58, 59).

Section 7

macrophage:mir-155

The loss of function of SOCS-1 is a common feature in HCC and the HBx-mediated upregulation of miR-155 is a contributing factor in HBV-HCC

Corrected: The dysregulation of the SOCS-1 function as a tumour suppressor is common in HCC pathogenesis

subtitle 7.5, second paragraph:

blocks p53 stimulated miR-34 expression in hepatocytes leading to the upregulation of macrophage-derived chemokine (CCL22) that stimulates regulatory T-cells (Tregs) that, in turn, block effector T-cells allowing HBV expression to increase 

Corrected: The HBx protein can repress p53 stimulated miR-34 in hepatocytes leading to an upregulation of macrophage-derived chemokine (CCL22) stimulated regulatory T-cells (Tregs). Tregs, in turn, can block effector T-cells thus allowing HBV expression to increase (25, 188).

Following are similar to World J Gastroenterol. Jun 28, 2014; 20(24): 7971-7978 in bold:

Subtitle 2, third paragraph

"(HBx) is known to play a pivotal role in the pathogenesis of viral induced HCC. HBx is a multifunctional protein of 17 kDa which modulates several cellular processes by direct or indirect interaction with a repertoire of host factors"

Corrected: The HBx protein plays an important role in the pathogenesis of viral induced HCC. This multifunctional 17 kDa protein can modulate several cellular processes directly or indirectly as a result of its interaction with the host genome.

"several cellular processes such as oxidative stress, DNA repair, signal transduction, transcription, protein degradation, cell cycle progression and apoptosis." 

Corrected:  can influence a number of cellular processes including oxidative stress, cell cycle controls, apoptosis, DNA repair, as well as signal transduction, transcription and protein degradation

Adaptive immune system-T-cell

The frequency of Tregs in the HCC tumor microenvironment was significantly higher than that in tumor-surrounding tissue and biopsy specimens from healthy livers

Corrected: This study also demonstrated that Treg levels in the HCC tumor microenvironment were significantly higher than in normal surrounding tissue

Following are similar to Oncogene volume 34, pages3085–3094(2015) in bold:

Subtitle 6.1, NK cells 

are expressed by cells undergoing stress triggered by events such as viral infection or cell transformation. Engagement of NKG2D on NK cells 

Corrected: that are expressed by cells as a result of viral infection or cell transformation. NK cells are able to kill an infected or abnormal cell as a result of the engagement of NKG2D with MICA/MICB on the targeted cell

Subtitle 6.1, B cell

control of B cell development in the bone marrow depends on the commitment of progenitor cells to the B cell lineage by the activation of transcription factor networks, V(D)J recombination events and selection for effective antigen receptors

Corrected: B-cell development in the bone marrow is controlled by the commitment of progenitor cells to the B-cell lineage as a result of the activation of transcription factor networks, as well as V(D)J recombination and the selection of antigen receptors.

miR-155-deficient B cells have defective antibody class switching and differentiation into plasma cells, resulting in an impaired humoral response to T cell-dependent antigenic

Corrected: B-cells that are miR-155-deficient can have a defective humoral response to T-cell-dependent antigenic stimulation because of an impaired antibody class switching and differentiation into plasma cells

T-cells

(TCR), as occurs during activation-induced cell death, is known to depend on the CD95–CD95 ligand pathway

Corrected: